# CHML promotes liver cancer metastasis by facilitating Rab14 recycle

Tian-Wei Chen[1,2,3], Fen-Fen Yin[1,2], Yan-Mei Yuan[1,2], Dong-Xian Guan[1], Erbin Zhang[1,2], Feng-Kun Zhang[1,2], Hao Jiang[1,2], Ning Ma[1,2], Jing-Jing Wang[1], Qian-Zhi Ni[1,2], Lin Qiu[1], Jing Feng[3], Xue-Li Zhang[3], Ying Bao[4], Kang Wang[5], Shu-Qun Cheng[5], Xiao-Fan Wang[6], Xiang Wang[7], Jing-Jing Li[1,2] & Dong Xie[1,2,8,9]

Metastasis-associated recurrence is the major cause of poor prognosis in hepatocellular carcinoma (HCC), however, the underlying mechanisms remain largely elusive. In this study, we report that expression of choroideremia-like (CHML) is increased in HCC, associated with poor survival, early recurrence and more satellite nodules in HCC patients. CHML promotes migration, invasion and metastasis of HCC cells, in a Rab14-dependent manner. Mechanism study reveals that CHML facilitates constant recycling of Rab14 by escorting Rab14 to the membrane. Furthermore, we identify several metastasis regulators as cargoes carried by Rab14-positive vesicles, including Mucin13 and CD44, which may contribute to metastasis-promoting effects of CHML. Altogether, our data establish CHML as a potential promoter of HCC metastasis, and the CHML-Rab14 axis may be a promising therapeutic target for HCC.

[1] CAS Key Laboratory of Nutrition, Metabolism and Food Safety, Shanghai Institute of Nutrition and Health, Shanghai Institutes for Biological Sciences, Chinese Academy of Sciences, Xuhui district 200031, China. [2] University of Chinese Academy of Sciences, Chinese Academy of Sciences, Xuhui district 200031, China. [3] Department of General Surgery, Fengxian Hospital Affiliated to Southern Medical University, 6600 Nanfeng Road, Shanghai 201499, China. [4] Department of Surgery, First People's Hospital Affiliated, Huzhou University, Huzhou 313000, China. [5] Department of Hepatic Surgery VI, Eastern Hepatobiliary Surgery Hospital, Second Military Medical University, Changhai Rd No. 225, Shanghai 200438, China. [6] Department of Pharmacology and Cancer Biology, Duke University Medical Center, Durham, NC 27710, USA. [7] Key Laboratory for Translational Medicine, First People's Hospital Affiliated, Huzhou University, Huzhou 313000, China. [8] NHC Key Laboratory of Food Safety Risk Assessment, China National Center for Food Safety Risk Assessment, Beijing 100022, China. [9] School of Life Science and Technology, ShanghaiTech University, 393 Middle Huaxia Road, Shanghai 201210, China. Correspondence and requests for materials should be addressed to X.W. (email: wangxiang004@163.com) or to J.-J.L. (email: tide7@163.com) or to D.X. (email: dxie@sibs.ac.cn)

Hepatocellular carcinoma (HCC) is one of the most common human malignancies in the world, ranking as the second leading cause of cancer-related death[1]. The high mortality is attributed to high rate of recurrence after hepatic resection (HR) and radiofrequency ablation (RFA)[2], with a 5-year recurrence rate exceeding 70%[3]. Since HCC recurrence is often predicated on the presence of vascular micro-invasion and micrometastases[4], investigating the underlying mechanism of HCC metastasis would provide therapeutic opportunities for this malignancy.

Aberrant signaling pathways play key roles in HCC metastasis[5], which was caused by genetic, epigenetic alternations[5], post-translational regulation, and localization[6]. Subcellular localization of proteins is mainly regulated by Rab GTPases[7,8], which comprises more than 60 members and regulates vesicles transport, proteins trafficking, membrane targeting[9]. Emerging studies have suggested the involvement of Rabs in HCC progression and metastasis. Significant correlation between Rab27A and Rab27B expression and tumor tumor-node-metastasis (TNM) classification was noted[10]. In another study, Rab25 was found to promote HCC cell proliferation and invasion[11].

Rab GTPases need to be geranylgeranylated on either one or two cysteine residues in their C-termini in order to localize to the correct intracellular membrane and be functional[12], and this process is accomplished by Rab escort protein (REP) and GGPP transferase (RabGGTase, RGGT)[12,13]. REP protein binds newly synthesized Rab proteins and presents them to the catalytic RGGT. REP family consists of only 2 members: choroideremia (CHM, REP1) and choroideremia-like (CHML, REP2). In recent reports, REP1 has been proved to promote progression of cervical, lung, and colorectal cancer[14], however, the function of CHML in cancer remains largely elusive.

In this study, we found that CHML expression was dysregulated in HCC and was closely correlated to clinicopathologic parameters. We further demonstrated that CHML regulated migration and invasion abilities of HCC cells through Rab14. CHML escorted Rab14 to membrane, which supported the constant recycling of Rab14. We also identified proteins carried by Rab14-positive vesicles as downstream effectors of CHML, such as Mucin13 and CD44, which were important regulators of HCC metastasis. Collectively, our study revealed the function and mechanism of CHML–Rab14 axis in HCC metastasis, providing a potential therapeutic target for HCC.

## Results

### Clinical significance of CHML expression in HCC.
To determine the expression pattern of CHML in HCC, we quantified the mRNA level of CHML in 45 paired HCC tissues and their matched normal counterparts by real-time PCR. The expression of β-actin was used as internal control. Upregulation of CHML was observed in 42 pairs, accounting for 93% of the total specimen examined (Fig. 1a). Moreover, in silico analysis of two independent datasets from oncomine (Chen Liver, $n = 179$ and Wurmbach Liver, $n = 45$) demonstrated that CHML mRNA levels were significantly higher in HCC compared to their normal counterparts (Fig. 1b), and this pattern could also be seen in TCGA dataset (TCGA, liver cancer data set) (Supplementary Fig. 1a). Consistently with the mRNA expression pattern, we also observed elevated protein level of CHML in HCC tissues compared with the matched normal tissues in 21 out of 24 paired samples by western blot (Fig. 1c). Higher expression of CHML protein in HCC was further confirmed by immunohistochemistry (Fig. 1d). Taken together, these data indicated that the expression of CHML was increased in HCC.

To explore the clinical significance of CHML in HCC, an HCC tissue microarray (TMA) containing 297 specimen was stained with CHML antibody and scored in a standard manner as described previously[15] (Supplementary Fig. 1b). We found that HCC patients with higher expression of CHML (with $H$-score $\geq$ 60, 214 patients) manifested a shorter overall survival (HR = 1.4, $P = 0.0073$) (Fig. 1e). Consistently, TCGA data analysis also demonstrated an association between higher CHML expression and shorter overall survival ($n = 291$, HR = 1.8, $P = 0.028$) (Fig. 1f). Further analysis of CHML expression and clinicopathologic parameters revealed that high CHML expression was associated with serious ascites and more satellite nodules (Table 1), and most importantly, earlier recurrence (Table 2). Thus we examined whether CHML expression was associated with recurrence-free survival in our TMA dataset and TCGA dataset. Expectedly, high CHML expression group had shorter recurrence-free survival compared with low CHML expression group in both cohorts (Fig. 1g, h). In conclusion, high expression of CHML was associated with poor survival and early recurrence in HCC patients.

### CHML regulated both in vitro migration and in vivo metastasis of HCC cells.
High rate of HCC recurrence following resection is primarily due to intrahepatic dissemination of the tumor via the intrahepatic vascular system. Presence of microvascular invasion (MVI) has been reported to be one of the most important risk factors related to postsurgery recurrence, thus we evaluated CHML expression in HCC patients with or without MVI in two independent datasets from GEO. We found that CHML expression was higher in patients with MVI than those without MVI in both datasets (Fig. 2a). HCC tends to invade the intrahepatic vasculature, especially the portal vein. Portal vein tumor thrombus (PVTT) arising from the invasion of HCC cells into the portal vein, is a special type of intrahepatic metastasis of HCC. Therefore we examine the involvement of CHML in PVTT formation by analyzing the expression of CHML in paired tumor adjacent normal tissues, primary HCC tissues and PVTT tissues. Analysis of GEO dataset (GSE74656) demonstrated that mRNA expression of CHML was lowest in non-cancerous tissues, relatively higher in the primary HCCs, and further increased in PVTT tissues (Fig. 2b). We confirmed this expression pattern of CHML in 4 paired tissues by western blot (Fig. 2c). These data indicated the association between elevated CHML expression and PVTT development, which was consistent with the correlation between CHML expression, satellite nodules, and recurrence.

Considering the association between CHML expression and metastasis in clinic, we overexpressed it in PLC/PRF/5 and YY-8103 cells (Fig. 2e), which showed relatively low expression of CHML (Fig. 2d), and examined the migration and invasion capacities of CHML-overexpressing cells. Results of Boyden chamber assay and transwell invasion assay demonstrated that overexpression of CHML significantly promoted both migration (~2-fold) and invasion abilities (~4-fold) of HCC cells (Fig. 2f, g).

In addition, we knocked down the endogenous CHML by two independent shRNAs in both CSQT-2 and LM3 cells, two HCC cells with relatively high level of CHML (Fig. 3a, c). CSQT-2 was a highly metastatic HCC cell line derived from a resected PVTT tissue established in our laboratory[16]. As shown in Fig. 3b, d, CHML knockdown (KD) significantly reduced the migration and invasion capabilities of HCC cells compared to the control cells (>50%, Fig. 3b, d). The inhibitory effect on migration and invasion of CHML KD could also be observed in YY-8103 and PLC/PRF/5 cells (Supplementary Fig. 2a–d). To exclude the potential influence of proliferation, we performed MTT assay, and found that CHML exerted little effect on growth rate of HCC

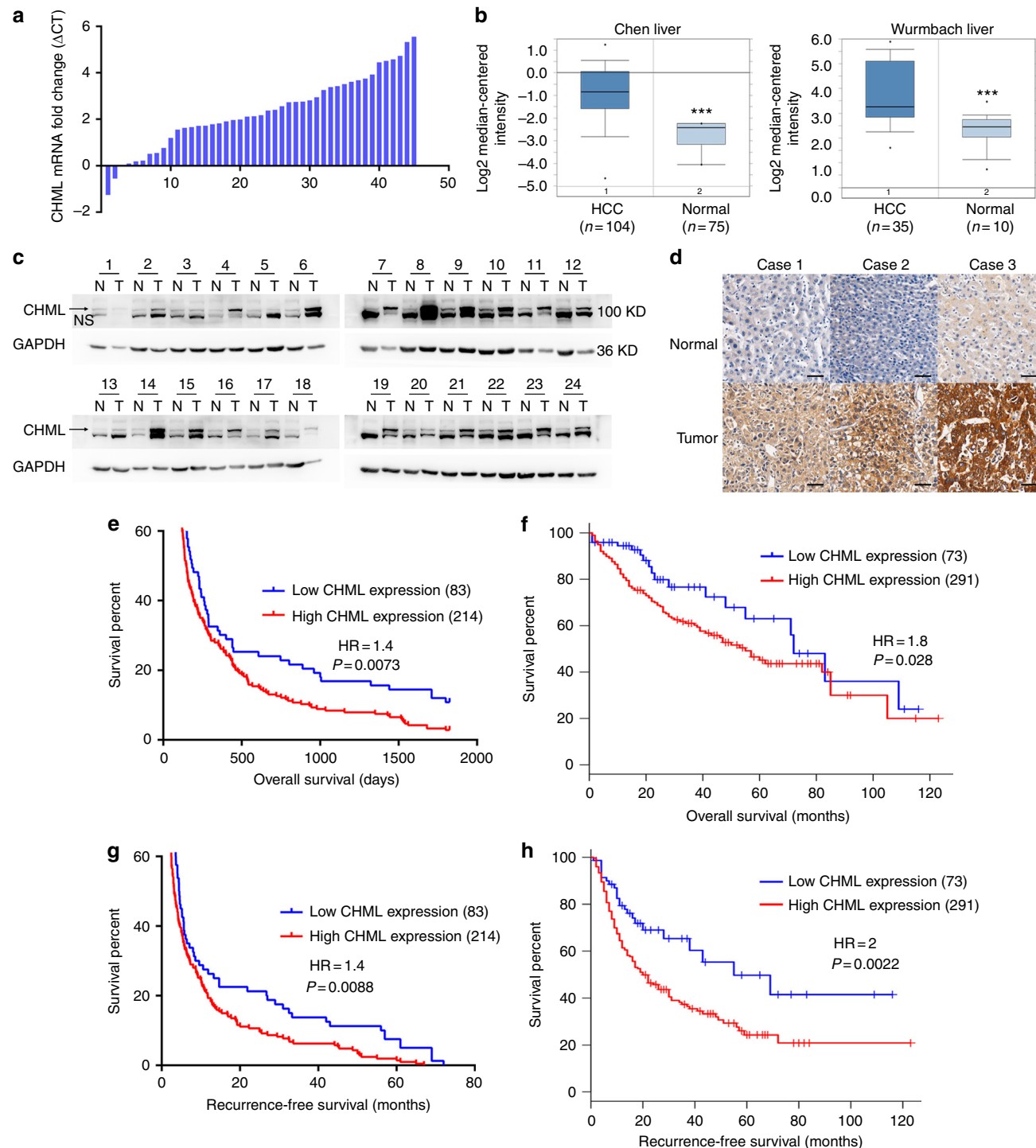

**Fig. 1** CHML is upregulated in HCC and predicts poor survival. **a** Real-time PCR analysis of CHML mRNA in 45 paired HCC tumor tissues and normal tissues. **b** Box plots comparing levels of CHML mRNA in normal human liver tissues and hepatocellular carcinoma tissues in published data sets from oncomine. ***$P < 0.001$, Student's $t$-test. **c** Western blot analysis of CHML expression in 24 paired normal (N) and HCC tumor (T) tissues. GAPDH is used as internal control. Arrow demonstrates the CHML band, and NS means non-specific band. **d** Immunohistochemistry staining of CHML in 3 paired normal liver and HCC tumor. Scale bar, 50 μm. **e** Kaplan–Meier analysis of overall survival of tissue microarray (TMA) data containing 297 patients. **f** Kaplan–Meier analysis of overall survival data from TCGA liver cancer data containing 364 patients. **g**, **h** Kaplan–Meier analysis of recurrence-free survival of the TMA data (**g**) and TCGA-LIHC data (**h**). All data are represented as mean ± s.e.m.

cells (Supplementary Fig. 4). Taken together, CHML positively regulated the migration and invasion of HCC cells in vitro.

The promoting effect on migration and invasion of HCC cells by CHML in vitro indicated its influence on HCC metastasis in vivo. To test this hypothesis, intrahepatic metastasis assay and

tail vein metastasis assay were performed. In intrahepatic injection assay, luciferase-labeled control and CHML KD CSQT-2 cells were orthotopically injected into the left hepatic lobes of nude mice ($n = 8$ mice in each group), respectively. Signal intensities in CHML KD group were much weaker than

**Table 1 Relationship between expression of CHML in HCC and clinicopathologic features**

| Characteristic | Low expression group ($n = 83$) | High expression group ($n = 214$) | $\chi^2$ | P value |
|---|---|---|---|---|
| Liver cirrhosis | | | | |
| No | 17 | 48 | 0.133 | 0.716 |
| Yes | 66 | 166 | | |
| Tumor size (cm) | | | | |
| ≤5 | 17 | 32 | 1.327 | 0.249 |
| >5 | 66 | 182 | | |
| Tumor capsulation | | | | |
| No | 26 | 48 | 2.53 | 0.112 |
| Yes | 57 | 166 | | |
| Ascity | | | | |
| No | 71 | 160 | 4.018 | 0.045* |
| Yes | 12 | 54 | | |
| Satellite nodules | | | | |
| No | 54 | 99 | 6.477 | 0.011* |
| Yes | 29 | 105 | | |
| Relapse | | | | |
| No | 23 | 48 | 0.917 | 0.338 |
| Yes | 60 | 166 | | |

*$P < 0.05$ values are set for significant difference

**Table 2 Relationship between CHML expression and patients' relapse time**

| Characteristic | Low expression group ($n = 60$) | High expression group ($n = 166$) | $\chi^2$ | P value |
|---|---|---|---|---|
| Relapse time | | | | |
| ≤10 | 50 | 154 | | 4.467 0.035* |
| >10 | 10 | 12 | | |

*$P < 0.05$ values are set for significant difference

those in control group at 8 weeks after injection (Fig. 3e), which was consistent with less tumor foci in the livers in CHML KD group (Fig. 3f). Similar phenotype was observed in the same model injected with LM3 and YY-8103 cells (Supplementary Fig. 2e, f). In the tail vein assay, CHML KD and control luci-CSQT-2 cells were injected into the tail veins of nude mice ($n = 8$ for each group) to examine their ability of lung colonization. Luciferase signals from the lungs of CHML KD group were significantly lower compared with control group at 8 weeks after injection (Fig. 3g), which was consistent with less foci in the lungs of the mice in CHML KD group (Fig. 3h). Similar results were also obtained with LM3 cells (Supplementary Fig. 2g). Consistently, in orthotopic xenograft model, while equal amount of tumor tissues derived from LM3 cells were bound to the livers of nude mice, less tumor foci in lungs were observed in shCHML group (Supplementary Fig. 3). Since high CHML expression predicted poor survival in clinic, we also evaluated the impact of CHML on survival of tumor-bearing mice in the intrahepatic injection model ($n = 10$ in control group and $n = 12$ in shCHML group). Kaplan–Meier survival plot revealed dramatically prolonged survival of the mice in shCHML group compared to control group ($P < 0.001$; Supplementary Fig. 3h). These results suggested that downregulation of CHML suppressed metastasis of HCC cells in vivo.

**Migration/invasion-promoting effect of CHML was mediated by Rab14.** Considering the critical role of REP protein in Rab geranylgeranylation[22] and the involvement of Rabs in metastasis, we firstly examined the expression of multiple Rab proteins in CHML knockdown cells. However, we found that total amount of some classical Rab proteins, including Rab1, Rab4, Rab5, Rab7, Rab11, Rab27, were not influenced by CHML knockdown (Supplementary Fig. 5a). Since geranylgeranylation is required for membrane targeting of Rabs, we performed Triton X-114 partition assay, which could distinguish unprocessed Rabs in aqueous phase from geranylgeranylated Rabs in Triton X-114 phase[17]. However, CHML knockdown did not influence the distribution of Rabs in Triton X-114 phase and aqueous phase, indicating that downregulation of CHML may not affect geranylgeranylation of Rab proteins (Supplementary Fig. 5b).

The above observations suggested that CHML may not generally influence expression and modification of Rab proteins. However, it might specifically affect certain Rab. To verify this hypothesis, firstly, we sought to identify the Rab protein interacting with CHML. We overexpressed 3xFlag-CHML in 293T cells and performed immunoprecipitation (IP) assay with anti-Flag beads. The silver stained gel showed a strong CHML band, and a significant band just beneath 25KD, which may contain several Rab proteins (Fig. 4a). This band was further analyzed by tandem mass spectrum. The result showed that abundant amount of Rab14, and a less content of Rab10, Rab4A, and Rab1A were included in this band (Fig. 4b). To verify the MS/MS data, we transfected 293T cells with Flag-CHML, and performed IP analysis. As expected, strong interaction between CHML and endogenous Rab14 was detected (Fig. 4c). In addition, Flag-Rab14 could also interact with endogenous CHML (Fig. 4d). Furthermore, interaction between endogenous CHML and Rab14 was detected in PLC/PRF/5, YY-8103 (Fig. 4e, f), CSQT-2 and LM3 cells (Supplementary Fig. 5c, d). To examine whether this interaction was direct or indirect, we conducted GST pull-down assay, using purified 6xHis-CHML, GST-Rab14, GST-Rab4A and GST-Rab7A. As shown in Fig. 4e, GST-Rab14 could bind to CHML, while Rab4A showed weak interaction with CHML, and Rab7A could not bind to CHML (Fig. 4e). We next asked which domain of CHML mediated its interaction with Rab14. As predicted by the software SMART, CHML mainly consisted of 2 GDI domains and 1 c-terminal domain (Fig. 4h). Accordingly, we fused the 3 domains with GST tag, respectively, and GST pull-down assay revealed that both GDI domains but not the c-terminal domain interacted with Rab14 (Fig. 4i). Collectively, CHML directly interacted with Rab14 via GDI domains.

To evaluate whether Rab14 was required for migration/invasion-promoting effect of CHML, we knocked out Rab14 expression in PLC/PRF/5 and YY8103 cells by CRISPR-Cas9 (Fig. 5a), then overexpressed CHML in these cells (Fig. 5a). We found that although CHML overexpression promoted cell migration in control cells, this effect was almost depleted in Rab14-deficient cells (Fig. 5b, c), suggesting that CHML function was dependent on Rab14.

**CHML regulated recycling of Rab14.** To elucidate the detailed mechanism, we firstly checked whether prenylation of Rab14 was impaired by CHML knockdown. As shown in Fig. S4, knockdown of CHML did not affect geranylgeranylation of Rab14. Then we examined the localization of Rab14 upon CHML KD by immunofluorescent staining. While Rab14 predominantly exhibited perinuclear localization in control cells, the distribution became more dispersed in CHML KD PLC/PRF/5 and YY8103 cells (Fig. 6a, b). As localization of Rab proteins was controlled by their

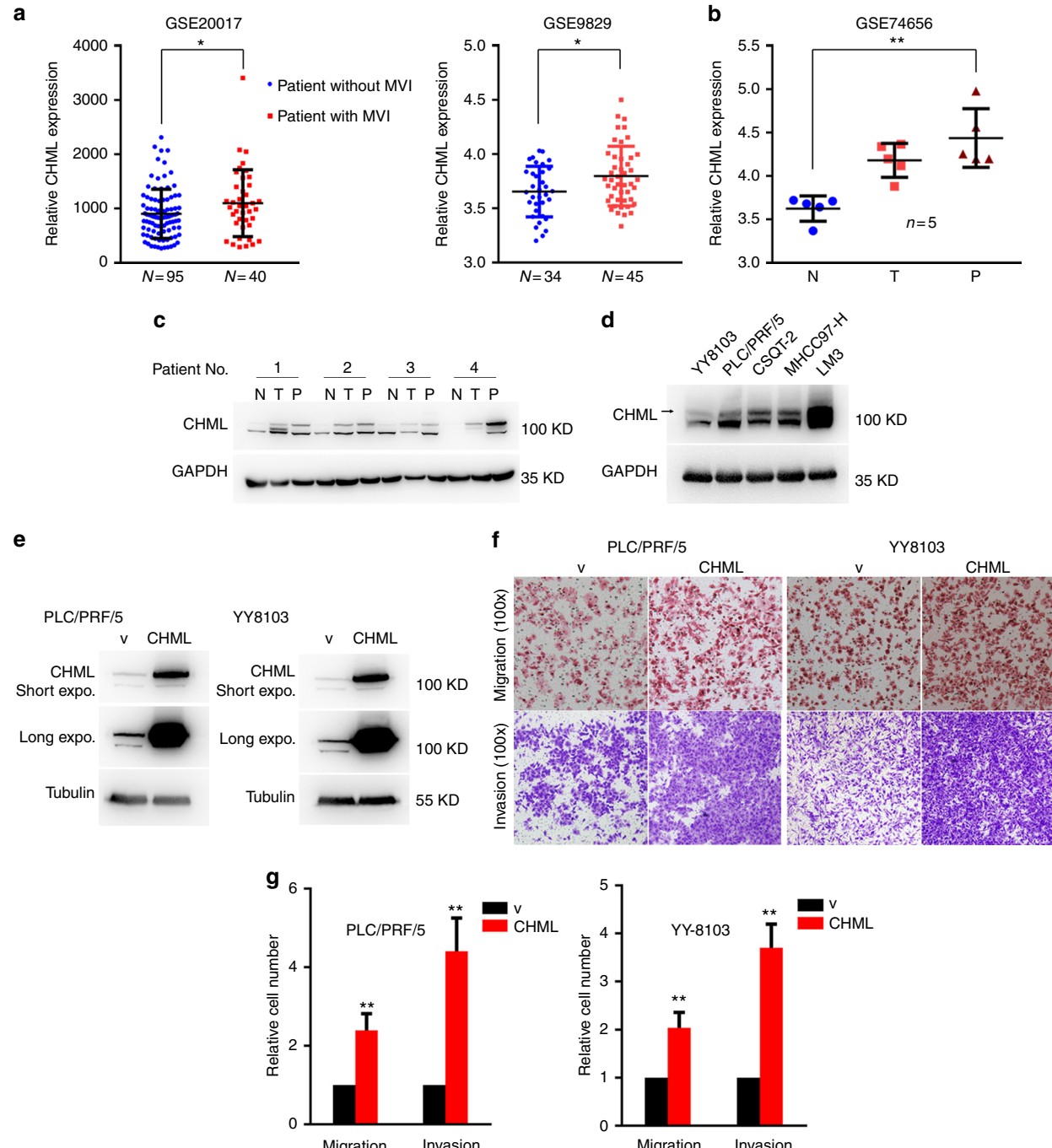

**Fig. 2** CHML overexpression promotes HCC cell metastasis and invasion. **a** Dot plots comparing CHML expression in patients with or without microvascular invasion (MVI) in two published datasets. *$P < 0.01$, Student's $t$-test. **b** Dot plot analysis of CHML expression in normal, HCC and portal vein tumor thrombus (PVTT) tissues. N normal, T tumor, P PVTT. **$P < 0.01$ by two-tailed unpaired Student's $t$-test. **c** Western blot analysis of CHML expression in normal, HCC, and PVTT tissues obtained from 4 patients. **d** Western blot analysis of CHML expression in 5 HCC cell lines. Arrow indicated the CHML band. **e** Western blot analysis of the overexpression efficiency of CHML in HCC cell line PLC/PRF/5 and YY8103. v vector control. **f** Boyden chamber and invasion assay are conducted to detect the migratory and invasive ability of control cells and CHML-overexpressed cells. Each representative image is shown. **g** Quantitative results are respectively illustrated for panel (**b**). Data are shown as the mean ± s.e.m., **$P < 0.01$ by two-tailed unpaired Student's $t$-test. All data are represented as mean ± s.e.m.

nucleotide binding status, we hypothesized GDP/GTP status of Rab14 might be affected by CHML. To validate this hypothesis, we purified GST-ΔRCP559-649 fusion protein, which was reported to specifically interact with GTP-Rab14[18]. GST pull-down assay using this substrate demonstrated that GTP-Rab14 was accumulated in CHML KD PLC/PRF/5 cells and YY8103 cells (Supplementary Fig. 6a, b), as well as CSQT-2 and LM3 cells

(Fig. 6c, d). GDP/GTP status of Rab14 was regulated by GDI, GAP and GEF, decreased amount of GDI and GAP could result in GTP accumulation (Fig. 6e). Thus CHML might function as a GDI or a GAP protein. Actually, GDI binds to GDP-loaded Rab while GAP binds to GTP-bound form. Based on this difference, we fused Flag tag to Rab14WT, and two mutants, Rab14S25N mimicking GDP-Rab14 and Rab14Q70L mimicking GTP-Rab14,

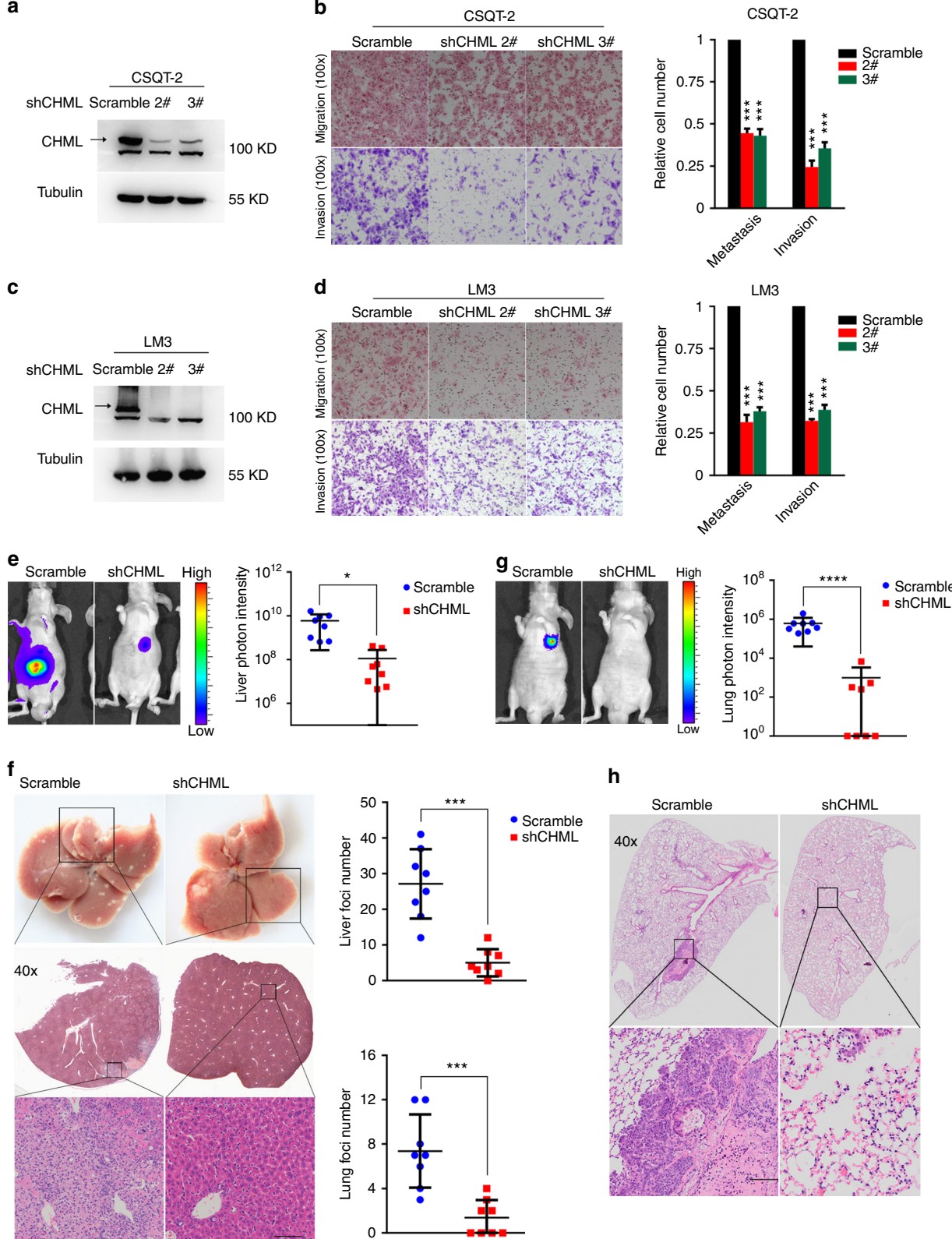

respectively, and performed immunoprecipitation assay. As shown in Fig. 6f, CHML preferentially bound to GDP-loaded Rab14S25N (Fig. 6f). Furthermore, immunofluorescent staining demonstrated that cellular CHML mainly colocalized with Rab14S25N-GFP fusion protein, rather than Rab14Q70L-GFP (Fig. 6g). These evidence revealed that CHML bound to Rab-GDP, suggesting its role as a GDI protein.

GDI functions to recycle Rab proteins[19], which included two processes: extraction of Rab from the target membrane and escorting Rab to the original membrane (Fig. 6h). Thus we examined which process CHML was involved. We isolated membrane of PLC/PRF/5 cells and used purified 6xHis-CHML and GST-GDI1 proteins for extraction assay. As shown in Fig. S6a, while Rab14 could be extracted by GDI1, Rab14 on the

**Fig. 3** Knockdown of CHML decreased cellular migratory and invasive abilities. **a**, **c** Western blot analysis of the knockdown efficiency of CHML in HCC cell line CSQT-2 (a) and LM3 (c). **b**, **d** Boyden chamber and invasion assay are conducted to detect the migration and invasion abilities of control cells and CHML-knockdown CSQT-2 cells (**b**) and LM3 cells (**d**). Each representative image is shown. Quantitative blot analysis was shown on the right. Data are shown as the mean ± s.e.m., ***P < 0.001 by two-tailed unpaired Student's t-test. **e** Left: representative images showing luciferase expression from intrahepatic tumors of both CSQT-2 control and shCHML group. Right: quantification of luciferase expression of intrahepatic tumors. *P < 0.01, Student's t-test. **f** Livers of both control and shCHML groups resected from intrahepatic metastasis mouse model. Tissues are photographed, fixed, and stained with hematoxylin and eosin (HE). Scale bar, 60 μm. ***P < 0.001 by two-tailed unpaired Student's t-test. **g** Left: representative images showing luciferase expression from lung metastasis of both CSQT-2 control and shCHML group. Right: quantification of luciferase expression of lung metastases. ****P < 0.0001 by two-tailed unpaired Student's t-test. **h** HE staining of lungs from both control and shCHML groups resected from tail vein injection metastasis mouse model. Scale bar, 100 μm. ***P < 0.001 by two-tailed unpaired Student's t-test. All data are represented as mean ± s.e.m.

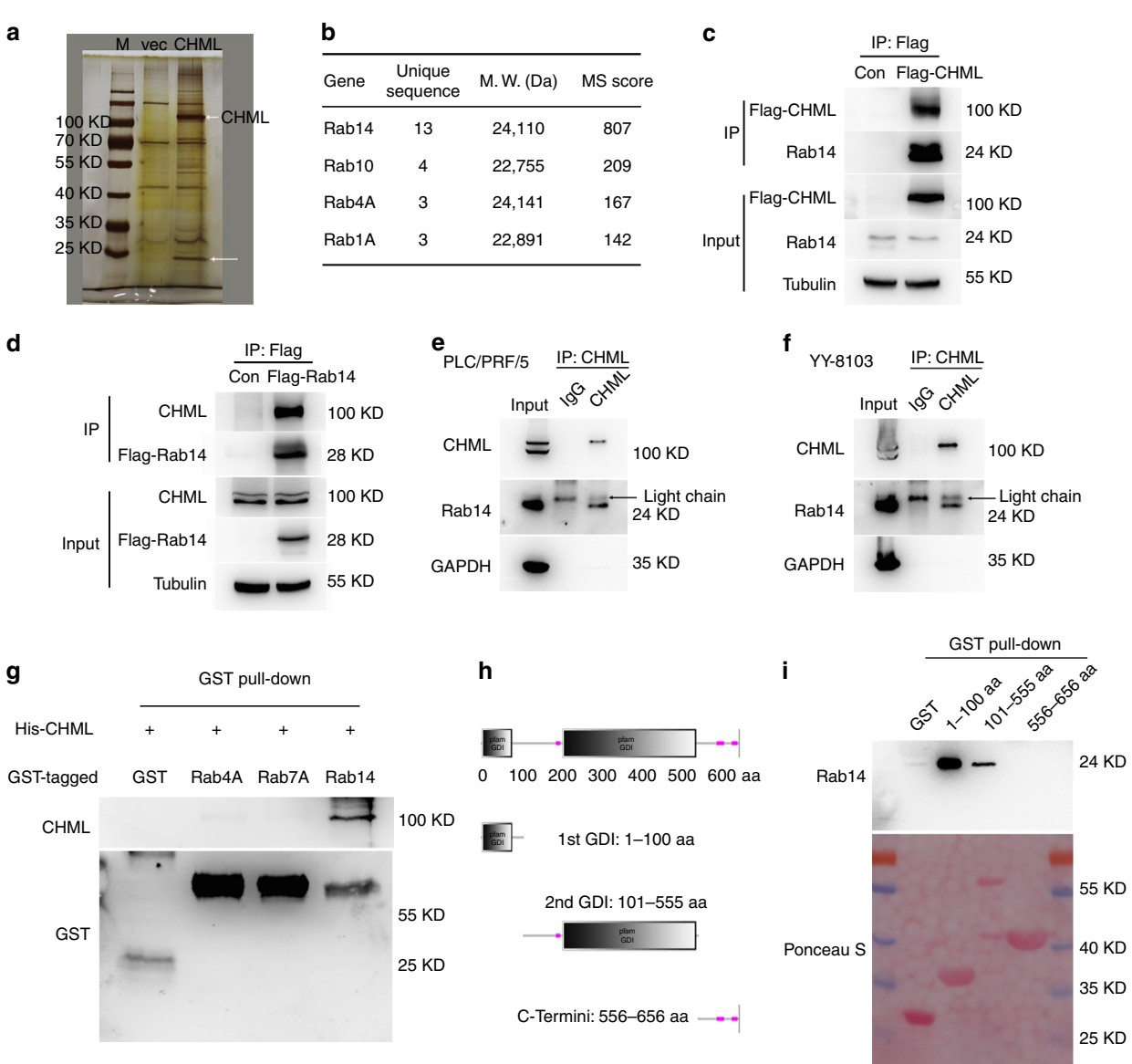

**Fig. 4** CHML interacts strongly with Rab14. **a** silver stained gel shows differential bands between control and CHML sample. White arrow indicates differential band, the upper one indicates Flag-CHML. M marker, vec vector control. **b** Tandem mass spectrum analysis of target band. **c**, **d** Interaction between CHML and Rab14 was demonstrated by immunoprecipitation. 293T cells were transfected with Flag-CHML (**c**) or Flag-Rab14 (**d**). Immunoprecipitation was performed at 36 h post transfection with Flag antibody. **e**, **f** Western blot results showing endogenous CHML interacted with Rab14 in PLC/PRF/5 (**e**) and YY-8103 (**f**) cells. Black arrow indicates light chain. **g** In vitro interaction between CHML and Rab14. Purified 6xHis CHML was incubated with indicated GST fusion proteins, GST pull-down assay was conducted. **h** Graphic shows the domains in CHML protein. **i** In vitro interaction assay showing the two GDI domains interacted with Rab14. The truncated domains of CHML were fused with GST, and incubated with Rab14, followed by GST pull-down and WB

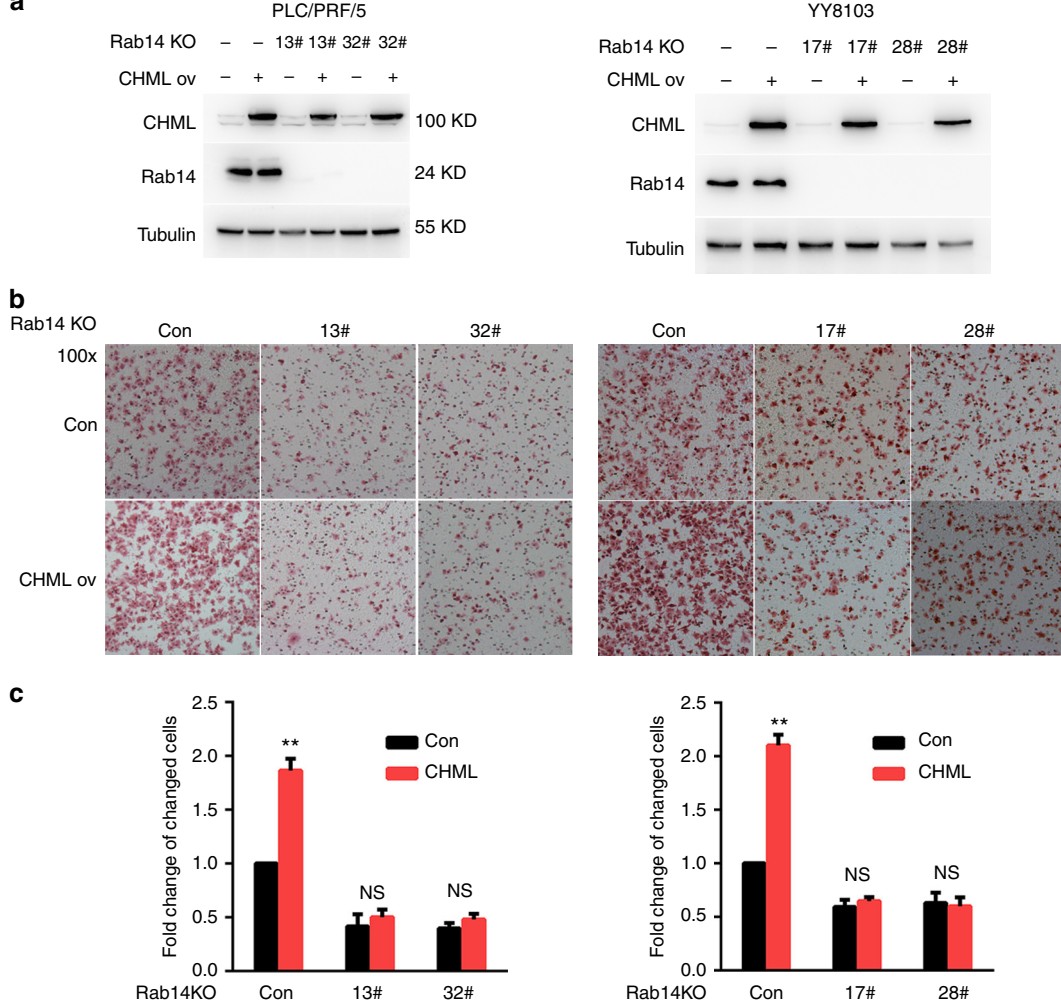

**Fig. 5** Function of CHML is Rab14-dependent. **a** Western blot analysis of the knockout efficiency of Rab14 and overexpression efficiency of CHML in HCC cell line PLC/PRF/5 and YY8103. **b** Boyden chamber assay was performed to detect the metastatic ability of these indicated cells. Representative images are shown. **c** Quantitative results are respectively illustrated for panel (**b**). NS not significant. Data are shown as the mean ± s.e.m., **$P < 0.01$ by two-tailed unpaired Student's $t$-test. All data are represented as mean ± s.e.m.

membrane or in the supernatant was not changed after incubation with purified CHML when compared to GST control (Supplementary Fig. 7a). To exclude the possibility that 6xHis-CHML purified from bacteria was inactive, we purified Flag-CHML from 293T cells and repeated the assay (Supplementary Fig. 7b). However, we obtained the same result, despite increasing amount of Flag-CHML (Fig. 6i), suggesting that CHML was not able to extract Rab14 from the membrane. Then we tested another possibility by examining membrane association of Rab14 in the presence of CHML. We isolated membrane from Rab14-depleted PLC/PRF/5 cells and purified CHML–Rab14 complex by immunoprecipitating Flag-CHML in 293T cells. Incubation of the Rab14-free membrane with CHML–Rab14 complex at 37 °C resulted in elevated level of Rab14 in membrane fraction in a dose-dependent manner (Fig. 6j). Therefore, CHML escorted Rab14 to membrane rather than extraction. In conclusion, CHML facilitated Rab14 recycling by escorting it to the original membrane.

**CHML regulated transport of metastasis regulators carried by Rab14-positive vesicle**. To clarify the role of Rab14 in the metastasis-promoting effect of CHML, we constructed two

Rab14 mutants deficient in GDP/GTP interchange to block the recycling of Rab14, GDP-locked mutant Rab14S25N and GTP-locked mutant Rab14Q70L. Then we stably expressed Rab14WT and the mutants in PLC/PRF/5 (Fig. 7a) and YY8103 cells (Fig. 7c), respectively. The Boyden chamber and invasion assay showed that Rab14WT cells enhanced cell migration and invasion compared with control cells, in contrast, both of Rab14S25N and Rab14Q70L decreased cellular migration and invasion (Fig. 7b, d). Therefore the balance between GDP-loaded and GTP-bound Rab was important to maintain a steady flow of Rab protein to exert its physiological function. This data also suggested that cargoes carried by Rab14-positive vesicle were required for HCC metastasis, which promoted us to identify them. To achieve this, we conducted vesicle immunoprecipitation assay (Fig. 7e). By sucrose density gradient ultracentrifugation and incubation with Rab14 antibody-coated Dynabeads 450, we isolated Rab14-containing endosome fraction from PLC/PRF/5 cells (Fig. 7f). Much weaker signal of Rab14 was detected in the reaction buffer after incubation with Rab14 antibody-coated beads compared with IgG beads, and stronger signal was observed in the precipitates with anti-Rab14 beads than IgG beads (Fig. 7g), indicating that Rab14-positive membrane fraction was successfully pulled down. Consistently, eluates from anti-Rab14 beads also

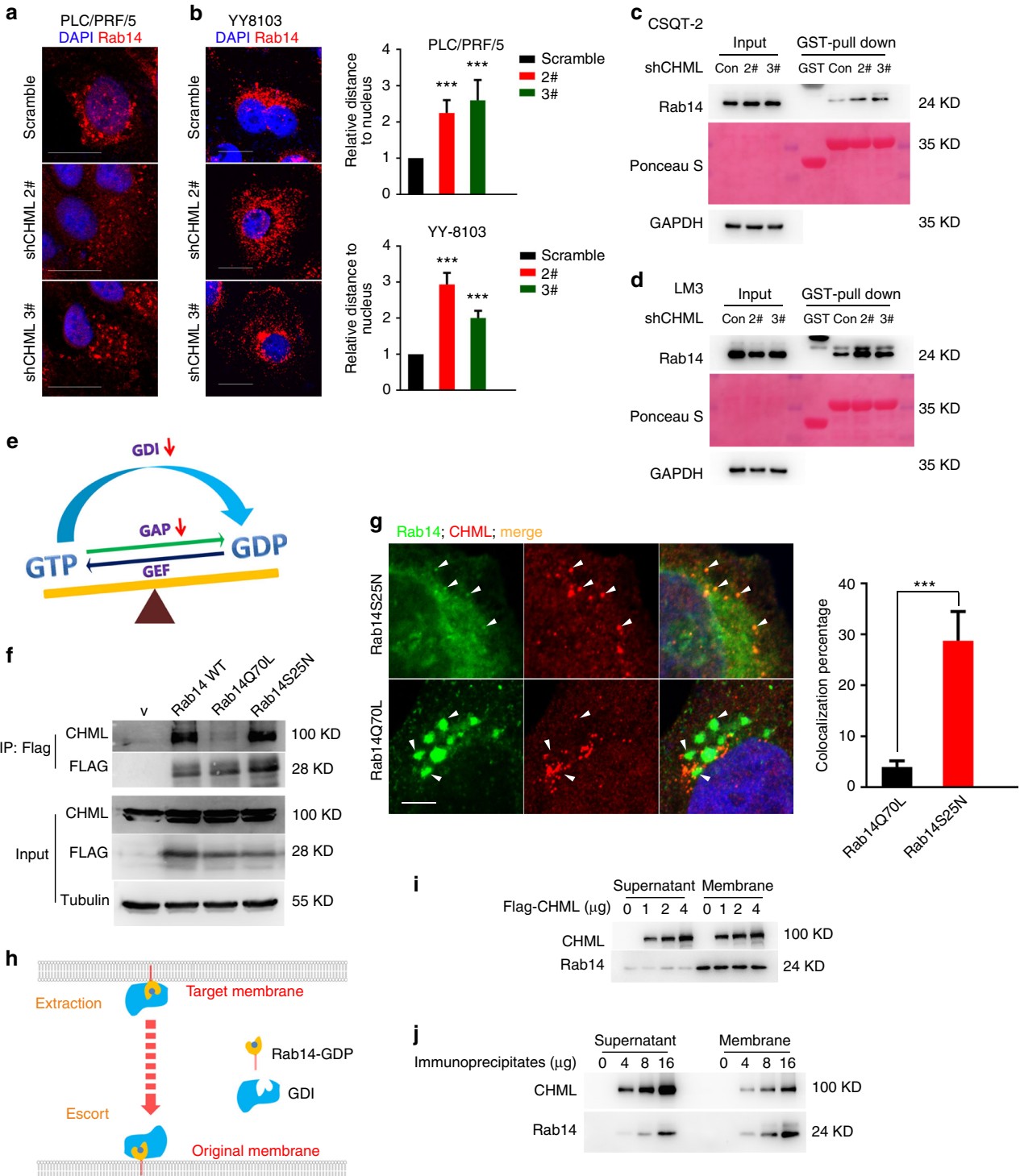

**Fig. 6** CHML serves to escort Rab14 to membrane. **a**, **b** Localization of endogenous Rab14 in PLC/PRF/5 (**a**) and YY8103 (**b**) cells was stained and representative images are shown. The distances of signals to nucleus were calculated and statistically presented. ***P < 0.001, Student's *t*-test. Scale bar, 20 μm. **c**, **d** Rab14-GTP in shCHML CSQT-2 (**c**) and LM3 (**d**) cells was pulled down by GST-ΔRCP559-649. Ponceau S indicates GST-ΔRCP559-649. **e** Schematic showing factors influencing Rab-GTP. **f** 293T cells were transfected with control vector and Rab14WT, Rab14Q70L, and Rab14S25N plasmid, respectively. After 48 h, immunoprecipitation assay was performed. **g** 293T cells were transfected with EGFP-Rab14S25N or EGFP-Rab14Q70L plasmid, and after 48 h endogenous CHML was stained. Images were photographed by a laser scanning microscope. Scale bar, 5 μm. White arrowheads indicated merged signal. The percentage of merged signal was counted and presented in bar plot. ***P < 0.001, Student's *t*-test. **h** Schematic showing the function of GDI. **i** PLC/PRF/5 Membrane fraction was incubated with Flag-CHML purified from 293T cells at 37 °C for 30 min, membrane Rab14 was detected by WB. **j** CHML immunoprecipitates from 293T cells were incubated with Rab14-depleted membrane, after incubation membrane fraction and supernatant were resolved by Western blot. All data are represented as mean ± s.e.m.

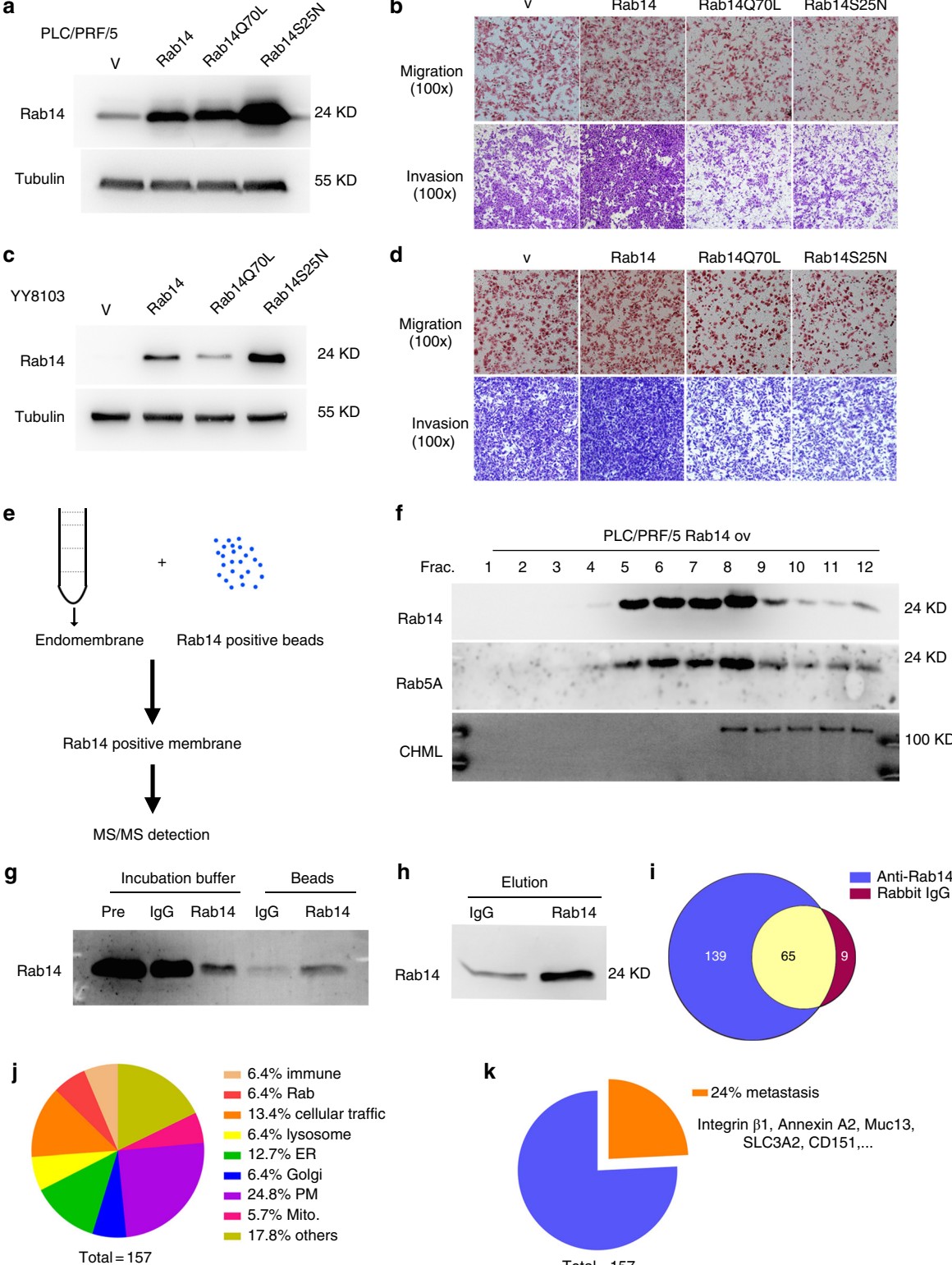

**Fig. 7** Rab14-positive membrane fraction contains metastasis-promotional factors. **a**, **c** Western blot analysis of the overexpression efficiency of Rab14WT and its mutants in HCC cell line PLC/PRF/5 and YY8103. V vector control. **b**, **d** Metastasis and invasion assays were conducted to detect these indicated cells. **e** Workflow to identify Rab14-positive vesicle components. **f** Sucrose-gradient centrifugation was performed to fractionate Rab14 containing endosomes. Rab5A was used as endosome marker. **g** Immunoprecipitation was performed using anti-Rab14 beads and endosomes. Incubation buffer-pre, endosome-containing buffer before incubating with beads. Incubation buffer-IgG or -Rab14, endosome-containing buffer after incubating with IgG beads or anti-Rab14 beads. **h** Western blot analysis of eluates from IgG and anti-Rab14 beads. **i** Venn diagram indicates proteins identified by MS/MS. **j** Venn diagram analysis of a total of 157 Rab14-positive membrane components. **k** 157 Rab14-positive membrane components were analyzed with their reported metastatic function in cancer

showed strong Rab14 signal compared to IgG beads (Fig. 7h), and they were further analyzed by tandem mass spectrum.

There were 204 proteins in anti-Rab14 eluate and 74 proteins in IgG eluate (listed in Supplementary Data 1). 65 proteins were overlapped, and 139 proteins were found in anti-Rab14 sample alone (Fig. 7i). A total of 157 proteins were significantly enriched in the Rab14-positive vesicles (Supplementary Data 2), which were further analyzed. Among them, Annexin A2, RUFY1, Rab10, GLUT2, and TfR were previously reported to be involved in the function of Rab14[20–22], which validated this assay. Function and localization analysis demonstrated that abundant cell membrane proteins (24.5% of total proteins) were contained in Rab14-positive vesicles, while the other proteins were widely distributed (Fig. 7j). 38 proteins have been reported to play a role in tumor metastasis (Supplementary Data 3), including CD44, integrin β1, Mucin13, SLC3A2, CD151, accounting for 24% of total proteins (Fig. 7k). Thus CHML and Rab14 might influence HCC metastasis through these metastasis regulators carried by Rab14-positive vesicles.

Since Mucin13 and CD44 were well-known central players in tumor metastasis, we chose the two molecules to clarify the role of Rab14-positive vesicle in metastasis regulated by CHML. To confirm that Mucin13 and CD44 were carried by Rab14-positive vesicles, we transfected HCC cell YY-8103 with EGFP-Rab14 and stained endogenous Mucin13 and CD44. As shown in Fig. 8a, Mucin13 and CD44 were partially colocalized with Rab14, indicating that part of them was in Rab14-positive vesicles. As CHML KD resulted in accumulation of Rab14-GTP, we next examined whether localization of Mucin13 and CD44 was influenced by CHML KD. Mature Mucin13 and CD44 were glycosylated and were presented on the cell surface, thus we examined whether cell surface fraction of these two molecules were altered. Result of cell surface protein biotinylation assay exhibited significant decrease in cell surface-associated Mucin13 and CD44 in CHML KD cells compared with control cells (Fig. 8b, c). To confirm this observation, we examined the expression of endogenous Mucin13 and CD44 in control and CHML KD cells by immunofluorescence. While obvious plasma membrane-localized CD44 were observed in control cells, CHML KD resulted in reduced membrane localization of CD44 (Fig. 8d). For Mucin13, plasma membrane localization was also significantly decreased in CHML KD cells compared with control cells (Fig. 8d). Thus, CHML KD decreased cell membrane-associated Mucin13 and CD44, which might in turn influence migration, invasion, and metastasis of HCC cells.

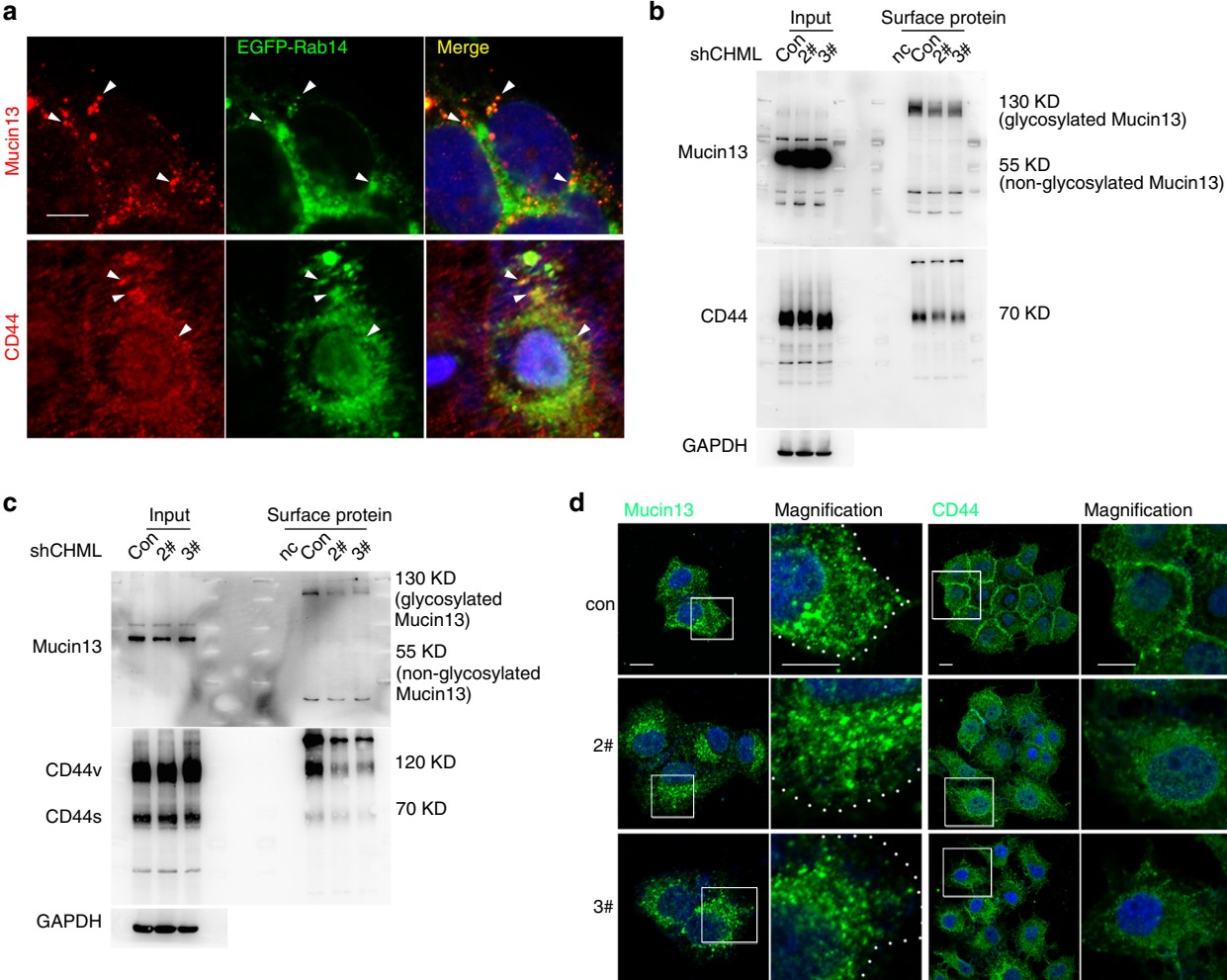

**Fig. 8** CHML KD influences membrane localization of CD44 and Mucin13. **a** Representative images showing Mucin13 and CD44 were partially co-localized with EGFP-Rab14 in CSQT-2 cells. Scale bar, 20 μm. White arrowheads indicated merge sites. **b**, **c** Western blot analysis of indicated surface protein in YY-8103 (**b**) and CSQT-2 cells (**c**). **d** Representative images showing endogenous distribution of CD44 and Mucin13 both in YY-8103 control and shCHML cells. Scale bar, 20 μm

## Discussion

CHML, and its REP homolog CHM, was responsible for the isoprenylation of Rab proteins[23]. Mutation of CHM causes degeneration of choroid and finally choroideremia in human. As for cancer, Yun et al. reported that oncogenic role of CHM was mediated by EGFR signaling recently[14]. CHML is considered to help mature Rabs like CHM[24], however, physiological function of CHML remains largely elusive, especially in cancer. In this study, we demonstrated that CHML was upregulated in HCC, and high CHML expression correlated significantly with more satellite nodules, earlier recurrence and poorer survival. Consistently, TCGA data also supported our observation that higher CHML expression in HCC patients was associated with shorter survival time. Further analysis revealed that HCCs with MVI expressed higher levels of CHML, and PVTT tissues expressed increased level of CHML compared to primary HCC tissues. Indeed, we found that knockdown of CHML decreased migration, invasion, and metastasis of HCC cells, while overexpression of CHML enhanced HCC cell migration and invasion, which was consistent with the clinical analysis.

Rab proteins act as key regulators of intracellular vesicular transport, and are assumed downstream targets of CHML[25–27]. There is mounting evidence of the involvement of Rab in cancer progression[28–30]. In our study, it was intriguing to find that intracellular CHML preferentially interact with Rab14, rather than other Rabs, such as Rab10 and Rab4A. Previous studies showed that CHML could present several Rab proteins, including Rab1A, Rab5A, Rab6 to geranylgeranyl transferase[31]. However, we could not detect interaction between CHML and these Rabs in 293T cells (Supplementary Fig. 8a). Actually, direct REP–Rab interaction is not a prerequisite for the geranylgeranylation of Rabs as demonstrated by Rudi et al. They revealed that in the alternative pathway of Rab geranylgeranylation, Rab substrates associate preferentially with pre-formed REP-RGGT complex, and unprenylated Rab5 as well as Rab11 could not form stable complex with REP1 both in vivo and in vitro[32]. This may explain why it is difficult to detect several Rabs–CHML interactions in the absence of RGGT in our system. In another study, Rak and colleagues[24] reported strong interaction between REP2 and Rab7, which seems inconsistent with our finding. However, different methods were employed in the two studies. Rak et al. performed fluorescence titration assay, which may be more sensitive to detect Rab–REP interaction than GST-pulldown assay. In fact, we determined the dissociation factor of Rab14 and Rab7A binding to CHML by fluorescence titration assay, respectively, and the results showed that Kd for Rab14–CHML interaction was ca. 0.36 nM, while Kd for Rab7A–CHML interaction was ca. 7.84 nM (Supplementary Fig. 8). Although CHML strongly interacted with Rab14, CHML knockdown did not affect the maturation of Rab14 (Supplementary Fig. 5b). Instead, CHML facilitated constant recycling of Rab14 by escorting it to original membrane. This finding distinguished CHML from CHM.

GDP and GTP-bound states of Rabs are regulated by GEF, GAP, and GDI[33–35]. Our data demonstrated accumulation of GTP-bound Rab14 in CHML knockdown HCC cells, which was the first evidence indicating that a REP protein could influence the balance between GTP-bound "on" and GDP-bound "off" states of Rab proteins. Different from other small monomeric GTPases, Rabs need recycling back to the original membrane to initiate next round of vesicle transport[7,36]. GDI acted as a recycling factor for many Rabs[19,37,38]. A previous study reported that CHM could extract and escort Rab5A to membrane, and promote endosome fusion[39], suggesting a GDI-like role of CHM. We found that CHML could escort Rab14 to membrane, and preferentially interacted with GDP-bound Rab14. However, it could not extract Rab14 from membrane. These data suggested the potential GDI-like function of CHML for Rab14, and also revealed the difference between CHML and CHM.

Recent studies have revealed the oncogenic role of Rab14 in several cancers[40–45], which is consistent with our observation. Small monomeric GTPases, like Ras and Rac1, are active when they are bound to GTP, and some studies reported that Rab-GTP was the functional type in cancer progression[46,47]. However, we found that migration/invasion-promoting role of Rab14 was dependent on balance between Ran14-GTP and Rab14-GDP, and shift to either form resulted in reduced migration/invasion of HCC cells, which improved our understanding of the metastasis-promoting effect of Rab14. Rab14 participated in various transport pathways, including protein transport from Golgi to endosome[48,49], plasma membrane protein presentation[50,51], GLUT4 presentation[52,53], phagocytosis[54,55], and endosome recycling[56,57]. Consistent with the role of Rab14 in vesicle transport and its involvement in HCC metastasis, we supposed that Rab14-positive vesicles may deliver metastasis regulators. Therefore we identified the cargo proteins by vesicle immuno-precipitation and tandem mass spectrum. Our results confirmed several cargo proteins in Rab14-pocitive vesicles reported previously, including Annexin A2[20], HLA-A[57], TfR[21], and RUFY1[22], which validated our analysis. Importantly, 38 cargo proteins have been reported to be directly involved in cancer metastasis. Furthermore, we demonstrated that CHML KD decreased the plasma membrane localization of Mucin13 and CD44, two important players in HCC metastasis[58,59]. Therefore CHML might promote cancer metastasis through metastasis regulators in Rab14-positive vesicles, and further studies were required to clarify the involvement of specific cargo proteins in the metastasis-promoting role of CHML–Rab14 axis.

In conclusion, our discovery illustrated the metastasis-enhancing function of CHML in HCC, which was mediated by vesicle transport of Rab14. Since CHML was overexpressed in more than 80% HCC tissues examined, activation of CHML–Rab14 axis might be a general oncogenic event in HCC, which provides a potential therapeutic target for HCC metastasis.

## Methods

**Materials.** Unless otherwise referred, all antibodies used in WB assay were diluted at 1:800. Rabbit polyclonal antibodies recognizing CHML (HPA029628), Rab14 (R0656) were purchased from Sigma-Aldrich. Anti-REP2 for endogenous immuno-precipitation was from Absci (AB41393). Anti-Rab1B (A7514), anti-Rab5A (A1180), anti-Rab5B (A7447), anti-Rab27A (A1934), anti-Rab31 (A7506) were from Abclonal. Anti-Rab4A (10347-1-AP), anti-Rab7A (55469-1-AP), anti-Flag (20543-1-AP) were from Proteintech. Anti-Rab4B (sc-376386), anti-Rab11A (sc-166912), anti-GAPDH (sc-47724), anti-α-tubulin (sc-73242), anti-β-actin (sc-58673) were from Santa Cruz. Anti-GST was from Sangon (D199985). Anti-Flag M2 (A2220) beads was from Sigma Aldrich. Glutathione Sepharose 4B, rProtein A sepharose, protein G were from GE. Dynabeads-M450 was from Thermo Fisher. Amicon ultra-0.5 and Amicon 10K ultra-15 were from Millipore.

**Cell culture and HCC samples.** HEK293T (SCSP-502) was purchased from Cell Bank of Type Culture Collection of Chinese Academy of Sciences, Shanghai Institute of Cell Biology, Chinese Academy of Sciences. YY-8103 and LM3 were from Zhong-Shan Hospital, Fudan University. PLC/PRF/5 was kindly provided by Shu-Qun Cheng at Eastern Hepatobiliary Surgery Hospital, Second Military Medical University, Shanghai, China. CSQT-2 was established in our lab. All cells were grown in DMEM (Gibco) supplemented with 10% fetal bovine serum (FBS; Anlite) and 1% penicillin/streptomycin (Sangon Biotech). All these cell lines were incubated at 37 °C in a humidified 5% $CO_2$ atmosphere. Human HCC samples were snap frozen in liquid nitrogen immediately after surgery. The protocol for tissue collection was approved by the Ethics Committee of Eastern Hepatobiliary Surgery Hospital, Second Military Medical University, Shanghai, China. Informed consent was written by all patients and all experiments were approved by the Ethical Committee.

**RNA isolation and real-time PCR.** Total RNA from HCC and paired normal tissues was isolated using TRIzol reagent (Invitrogen). 2 µg of total RNA was transcribed to cDNA with the reverse transcription kit (Promega, Madison). Real-time PCR was performed using SYBR premix Taq (Yeasen Biotech) in

Mx3000P Real-time detection system (Stratagene). Primers used are listed in Supplementary Table 1.

**Immunohistochemistry (IHC) and immunofluorescence (IF)**. Anti-CHML (Sigma-Aldrich) was diluted 1:20 to stain HCC TMA slides and 1:50 for immunofluorescence staining. Anti-Rab14 (Sigma) was diluted 1:100 in IF assay. Briefly, for IHC, sections of clinical specimens were deparaffinized with xylene and rehydrated with ethanol, followed by staining with CHML antibody and horseradish peroxidase (HRP)-linked anti-Rabbit IgG, and further development with 3,3′-diaminobenzidine (DAB). TMA sections were further photographed and analyzed by Vecture 2 (PerkinElmer). The same algorithm was used to score every core (Supplemental Materials). For IF, cells on slides were fixed in 4% formaldehyde, followed by staining with indicated primary antibodies and fluorescent secondary antibody (Alexa Fluor 488-conjugated donkey anti-Rabbit IgG and Alexa Fluor 555-conjugated goat anti-Rabbit IgG, 1:1000) and were photographed by confocal microscope (Zeiss, LSM 510 NLO).

**Plasmids and stable cell lines**. ORFs encoding CHML, GDI1, and all Rabs were cloned from HCC tissue samples. Primers are listed in Supplementary Table 1. For transient expression, all these ORFs were cloned into p3xFlag-CMV-10 vector. pEGFPC1 is used for Rab14, Rab14S25N, and Rab14Q70L fusion with EGFP. pHAGE-fEF1a-IRES-ZsGreen was used for CDS lentivirus production. pLKO.1 was used to produce shRNA lentivirus. LentiCRISPRv2 was used to produce Cas9-mediated gene knockout lentivirus. For stable cell lines production, HCC cell lines were transfected with lentivirus for 48 h, followed by GFP sorting (pHAGE-fEF1a-IRES-ZsGreen vector) or puromycin treatment (pLKO.1 vector; 3 days and longer).

**Western blot and immunoprecipitation**. 3xFlag tagged protein immunoprecipitation was performed according to the anti-Flag M2 manual. Briefly, HEK293T cells were washed twice by ice-cold PBS, and lysed in lysis buffer (50 mM Tris–HCl, pH = 7.4 with 150 mM NaCl, 1 mM EDTA, and 1% Triton X-100) with protease inhibitors for 10 min on ice, after centrifugation at $20,000 \times g$ for 15 min at 4 °C. Supernatants were incubated with anti-Flag M2 beads on a rotator overnight in cold room. After incubation, the beads were pelleted and washed by TBS (50 mM Tris–HCl, 150 mM NaCl, pH = 7.4) for 5 times. Elute with 3xFlag peptides for 1 h. The eluate was resolved by SDS-PAGE western blot.

**Boyden chamber and transwell assay**. Chemotactic cell migration was performed using a 12-well Boyden chamber. Briefly, the coarse side of polycarbonate film was coated with 50 μg/mL rat tail collagen type I at 4 °C overnight. 100 μL DMEM containing $1 \times 10^5$ cells were plated on the upper side. DMEM with 10% FBS was added in the lower inserts. The chamber was then incubated at 37 °C for 5–7 h. Cells that did not migrate through the pores of the film were manually removed by a rubber swab. Cells that migrated to the coarse side of the film were stained with eosin and photographed using an inverted microscope. Invasion assay was conducted in 24-well inserts (Corning Inc., Corning, NY, USA). Briefly, wells were laid with diluted matrigel (Corning; diluted by FBS-free DMEM) on ice, then incubating at 37 °C until the matrigel became concrete. Cells ($1 \times 10^6$) were seeded on the top and complete medium was added to the bottom side. After incubation for 48 h, non-invasive cells were removed from the upper side of the insert with a cotton swab. The bottom cells (invasive cells) were fixed with 4% paraformaldehyde for 20 min, stained with a 0.1% crystal violet solution for 30 min, and photographed using microscope. Numbers of cells were counted, and data were presented as the means of three randomly selected fields.

**Animal studies**. Animal studies were compiled with ethical regulations for animal testing and research and were approved by and performed in accordance with Institutional Animal Care and Use Committee of Shanghai Institutes for Nutritional Sciences, Chinese Academy of Sciences (SIBS-2018-XD-3).

For intrahepatic metastasis assay, cells were injected into the left lobes of livers of nude mice (6-week old mice; with $5 \times 10^5$ cells per mouse; Slaccas). 8 weeks post injection, both livers and lungs were photographed and foci numbers on the surface of these organs were counted followed by standard H&E procedure. Liver and lung foci were counted by microscope. For weekly luciferase reporter assay, mice were intraperitoneal injected with D-luciferin. They were anesthetized by isoflurane and photographed in IVIS imaging system (Xenogen).

For lung metastasis, cells ($4 \times 10^5$) were injected into tail vein of each nude mouse. 8 weeks post injection, all mice were sacrificed and both lungs and livers were subjected to procedures described above.

For orthotopic xenograft experiment, control or shCHML LM3 cells were injected into nude mouse subcutaneously, after 20 days, tumor tissues were resected, weighted, and transplanted to the livers of nude mouse. 45 days post transplantation, both livers and lungs were collected, followed by HE staining.

For survival assay, mice injected cells were fed normally until their death, date of death were recorded. And the recording time spanned 80 days.

**Triton X-114 partition**. Unprocessed Rab proteins and geranylgeranylated Rab proteins were separated by Triton X-114 partition method. Briefly, cells were

washed twice with ice-cold PBS and lysed in lysis buffer (50 mM Tris–HCl, pH = 7.4 with 150 mM NaCl, 5 mM MgCl, 1 mM dithiothreitol and 1% Triton X-114) with protease inhibitors for 10 min on ice. Lysates were centrifuged at $20,000 \times g$ at 4 °C for 20 min. The supernatant was incubated at 37 °C for 2 min, the cloudy state supernatant was centrifuged at $500 \times g$ for 4 min at room temperature. The upper aqueous phase and the lower Triton X-114 phase were collected respectively. The aqueous phase was added with Triton X-114 to a final concentration of 1% and Triton X-114 phase was added with lysis buffer containing no Triton X-114. Both samples were incubated in ice for 5 min until they became clear. Then they were warmed at 37 °C, and both phases were collected. Repeat for 3 times. The aqueous phase contained unprocessed Rab proteins and the Triton X-114 phase contained geranylgeranylated Rab proteins. Protein concentration was determined by Bradford method. They were further subjected to western blot.

**GST protein purification**. GST-Rab4A, Rab7A, Rab14, GDI1, ΔRCP559-649 were purified from BL21 according to the manual. Briefly, coding sequences were subcloned into pGEX-4T-1 vector. Vectors were transformed into BL21 bacteria. Grow the bacteria in LB medium with Ampicillin to A$_{600}$ 0.6–1. IPTG was added to a final concentration of 0.1 mM, continuing incubation for additional 6 h at 30 °C. Cells were pelleted down, washed twice, and resuspended in PBS with protease inhibitor. Sonicate for 30 min, add Triton X-100 to a final concentration of 1%. Shaking the sonicates at room temperature for 30 min followed by centrifuging at $16,000 \times g$ for 15 min at 4 °C. The supernatant was incubated with GST beads (GE) for 1 h at 4 °C. The beads were pelleted down, washed for 5 times with PBS. Eluting the beads with 10 mM reduced glutathione. Condense the purified protein by Amicon Ultra-10K membrane (Millipore) according to the manufacturer. Bradford reagent was used to determine the concentration of the purified protein.

**6xHis-CHML purification**. Coding sequence for CHML was cloned into pET28a vector. The purification procedure was the same with GST protein purification despite some minor differences. Briefly, cells were resuspended in PBS containing 5 mM imidazole before sonicating. Ni-NTA beads were used instead of GST beads, and the elution buffer containing 300 mM imidazole instead of 10 mM reduced glutathione.

**In vitro interaction assay**. 2 μg 6xHis-CHML was incubated with 2 μg GST or GST-Rab4A, Rab7A, Rab14 in binding buffer (20 mM Tris–HCl pH = 7.4 with 100 mM NaCl, 1 mM EDTA, 1 mM DTT, 10 mM MgCl$_2$, and 1% Triton X-100) for 4 h, then GST beads were added into the reaction buffer, incubating for another 2 h. Beads were pelleted down and dissolved by SDS-PAGE WB.

**Rab14-GTP detection**. GTP-Rab14 was detected by GST-ΔRCP559-649 pull down. Briefly, cells were plated onto the plate at the density of $2 \times 10^6$ cells per 10 cm dish. Harvested at 60–70% confluence, cells were washed twice with pre-cold PBS and lysed with 750 μL of ice-cold lysis buffer (25 mM Tris, pH 7.4, 1 mM EDTA, 5 mM MgCl$_2$, 1 mM DTT, 0.1 mM EGTA, 100 mM NaCl, 1% Nonidet P-40) with protease inhibitor (Sigma-Aldrich) on ice and then incubated with GST-ΔRCP559-649 fusion protein that had been adsorbed to glutathione agarose beads for 30 min. After that, the beads were washed 3 times with 1 mL of cold lysis buffer then resuspended in 40 μL of reducing electrophoresis sample buffer (2% SDS, 10% glycerol, 80 mM Tris, pH 6.8, 2 mM EDTA, 100 mM DTT, and 0.1% bromophenol blue) and analyzed by SDS-PAGE. After electrophoresis, samples were transferred to nitrocellulose membrane and immunoblotted with primary Rab14 antibody at a dilution of 1:1000.

**Cellular membrane preparation**. $10^8$ cells were washed 3 times with pre-cold PBS, and scraped with a rubber policeman. The cells were centrifuged at $800 \times g$ for 3 min, the pellet was resuspended in 5 mL homogenization buffer (250 mM sucrose buffered to pH 7.4 with 3 mM imidazole) and centrifuged again at $800 \times g$ for 3 min. The pellet was then resuspended in 1 mL of homogenization buffer with protease inhibitor and subjected to 30 tight strokes in Dounce homogenizer on ice. The homogenate was centrifuged at $3000 \times g$ at 4 °C for 10 min, the supernatant was repeated centrifugation again to thoroughly remove intact cells and nuclei. The supernatant was post nuclear supernatant (PNS). PNS was ultracentrifuged at $200,000 \times g$ at 4 °C for 1 h. Supernatant (cytosol) and pellet (membrane) were both collected.

**Extraction assay**. For extraction of Rab14 from membrane, membrane fraction dissolved in HEPES buffer (25 mM HEPES, 150 mM NaCl, pH = 7.4) was incubated with indicated proteins at 37 °C for 30 min followed by ultracentrifugation at $200,000 \times g$ at 4 °C for 1 h. The supernatant contained extracted Rab proteins and was further analyzed by WB.

**Escort assay**. Flag-CHML complex was purified from 293T cells as described in immunoprecipitation step. The 3xFlag eluate was condensed by Amicon Ultra-0.5. Protein concentration was determined by Bradford reagent. Membranes from Rab14-KO PLC/PRF/5 cells were incubated with gradient complex at 37 °C for

30 min, subjected to ultracentrifugation at $200,000 \times g$ at 4 °C for 1 h. The pellet was collected and further analyzed by WB.

**Rab14-positive endomembrane immunoprecipitation**. Anti-Rab14 (Sigma-Aldrich) antibody or the control Rabbit IgG was ligated to Dynabeads M450 (Thermo Fisher) according to the manufacturer's protocol. Briefly, 200 μg goat anti-Rabbit antibody was incubated with 1 mL beads in 1 mL 0.1 M borate buffer for 30 min, then added 20 μL 5% BSA, rotating at room temperature for 48 h. 200 μL beads were continued to incubate with 10 μg Rabbit anti-Rab14 antibody or control Rabbit IgG at 4 °C overnight. Endomembrane was prepared as described previously with some modifications[60]. Briefly, $3 \times 10^8$ PLC/PRF/5 cells were washed by pre-cold PBS, scrapped in 1 mL cold PBS, then incubated with hypotonic buffer (10 mM Tris–HCl) for 2 min, the pelleted cells were then resuspended in homogenesis buffer (250 mM sucrose, 3 mM EDTA) with protease inhibitor. Twenty tightly strokes of homogenizer were performed, followed by $3000 \times g$ centrifugation for 10 min. The supernatant was mixed with 62% sucrose to reach a final sucrose concentration of 42%. Discontinuous sucrose gradient ultracentrifugation was conducted, sucrose gradient were 42%, 35%, 25%, 4 mL for every gradient. Then centrifuge at $200,000 \times g$ overnight. Take 1 mL every time from top to bottom. Mix the 35% sucrose part, diluted 5 times with PBS. This fraction was then incubated with the previously prepared beads overnight. For MS/MS detection, beads were eluted by 1% SDS PBS for 5 min.

**Cell surface protein detection**. Cells in 10-cm dishes were incubated on ice for 3 min, washed by pre-cold PBS for 3 times, followed by incubation with 0.5 mg/mL sulfo-NHS-SS-Biotin (APExBio) for 10 min on ice, quenched by 192 mM Glycine solution, washed by ice-cold PBS for 3 times. Cells were lysed in IP lysis buffer, and the same amount of protein was incubated with streptavidin beads overnight (Thermo Fisher). Beads were washed and subjected to WB procedure.

**Fluorescent image quantification**. We quantified the distance of Rab14 signal to nucleus in ImageJ according to the method described previously. Briefly, different color channels were split (green for Rab14, blue for nucleus), followed by binary transformation, and then brightness-weighted average of the x and y coordinates all pixels (center of mass), for distance of pixel i $(Xi, Yi)$ to nucleus $(Xn, Yn)$, we used formula as below:

$$\sqrt{(Xi - Xn)^2 + (Yi - Yn)^2} \qquad (1)$$

We then calculated average distance of Rab14 signal to nucleus.

Colocalization analysis was performed in ZEN software. The analysis procedure was performed according to the tutorial.

**Statistical analysis**. All data are presented as the mean ± standard errors of the mean (s.e.m.). The Student $t$ test was used for the comparison of measurable variants of two groups. Survival curves were calculated using the Kaplan–Meier method, and differences were assessed by a log-rank test. Criterion for significance was $P < 0.05$ for all comparisons.

**Reporting summary**. Further information on research design is available in the Nature Research Reporting Summary linked to this article.

## Data availability

The source data underlying all figures as well as supplementary figures are provided as a Source Data file. All the other data supporting the findings of this study are available within the article and its Supplementary Information files and from the corresponding author upon reasonable request. A reporting summary for this article is available as a Supplementary Information file.

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

## Acknowledgements

This work was supported by the National Key R&D Program of China (2018YFC1604404 and 2018YFC1603002), the "Personalized Medicines—Molecular Signature-based Drug Discovery and Development", Strategic Priority Research Program of the Chinese Academy of Sciences (Grant No. XDA12010316), and National Natural Science Foundation of China (31520103907 and 81730083) to Dong Xie; National Natural Science Foundation of China (31771538), Youth Innovation Promotion Association of Chinese Academy of Sciences fund, and Sanofi-SIBS 2018 Young Faculty Award to Jing-Jing Li; and National Natural Science Foundation of China (31601143) to Dong-Xian Guan. The authors thank the New World Group for their Charitable Foundation to establish the Institute for Nutritional Sciences, SIBS, CAS-New World Joint Laboratory, which have given full support to this study.

## Author contributions

T.-W.C. and D.X. conceived this project. T.-W.C. and J.-J.L. designed the study. T.-W.C. performed most of the experiments and analyzed the data. F.-F.Y. and Y.-M.Y. constructed some vectors. D.-X.G., E.Z., F.-K.Z., H.J., N.M., J.-J.W., Q.-Z.N., L.Q., X.W., Y.B., X.-L.Z., J.F. and X.-F.W. provided essential advisement. K.W. and S.-Q.C. collected clinical samples. T.-W.C. wrote the manuscript.

## Additional information

**Competing interests:** The authors declare no competing interests.

