## [Peer Review File · Nature Communications]

Reviewers' comments:

Reviewer #1 (Expertise: Rab14, cancer, Remarks to the Author):

This manuscript describes a relationship between the choroideremia like (CHML) protein and the progression of hepatocellular carcinoma. The authors find that CHML expression is correlated with migration, invasion, and metastasis of hepatocellular carcinoma and that expression is correlated with survival in patients. Further results show that CHML interacts directly with Rab14 and promotes its delivery into membranes.

While the authors demonstrate that CHML is increased in HCC, they conclude that it plays a tumor promoting role. However, the increased expression may be well downstream of the actual carcinogenic event and may be a consequence of the transformation. Further, the importance of Rab14 in migration and cancer is established in the literature.

With respect to the cell biology experiments, I have a few concerns. To address the role of CHML in migration, it was overexpressed or knocked down in a HCC cell line which is therefore already transformed. It is not clear why endogenous levels in Figure 2 are so low while they are high in Figure 3. With respect to the migration/invasion assays, it is not clear why controls in Figure 2 migrate so much less than the controls in Figure 3. Also, data presentation is confusing. What is "fold of changed cells" or "percentage of changed cells" in the migration/invasion assay?

The co-immunoprecipitation and pull down assays provide strong evidence for an interaction between CHML and Rab14, but do not provide any insight into the domains that might mediate this interaction.

Further experiments test the role of CHML in Rab14 localization. While the authors state that knockdown of CHML results in a "more dispersed" distribution of Rab14 after CHML KO, the imaging presented does not support this conclusion. This is also true of Figure 6g. These data require careful quantification.

Overall, the imaging in Figure 6 is of low quality.

Final experiments test the types of proteins present in Rab14 positive vesicles and conclude that the presence of metastasis associated proteins provides a potential mechanism for CHML's role in cancer. However, these data highly preliminary and do not provide substantial insight. In fact, with Rab14's known localizations and function, it would be surprising if these proteins were not present.

The manuscript needs to be carefully edited for English.

Reviewer #2 (Expertise: Liver cancer, metastasis, Remarks to the Author):

In this manuscript, the authors reported that choroideremia-like (CHML) was up-regulated in HCC, and was associated with poor survival of HCC patients. Overexpression of CHML promoted the migration and invasion abilities of HCC cells.

Mechanically, CHML interacted with Rab14, and Rab14-positive vesicles contain several metastasis regulators, which might contribute to metastasis-promoting effects of CHML. Overall, this work is well executed and the results are interest to the field. However, there are some concerns needed to be further explained for better understanding.

Major concerns:

- The author concluded that choroideremia-like (CHML) is up-regulated in HCC and is closely associated with the metastasis of HCC. However, the data in Table 2 and Fig 1 is not enough to support the conclusion that “ high expression of CHML predict poor survival and earlier recurrence of HCC”. In addition to overall survival (OS), the association of CHML expression with the recurrence probability and time to relapse after HCC resection should be evaluated.
- The reason that 7404 and YY8103 HCC cell lines which are not commonly used were selected in this study should be explained. Besides the data of up-regulating and then knocking down the expression of CHML in 7404 and YY8103 cells , the results should be further validated in other HCC cell lines with high basic CHML levels and higher metastatic potentials .
- Besides the effect of overexpression of CHML on cell migration and invasion of HCC cells, does CHML affect the proliferation of HCC cells?

- In the figure 3d, the image present is not clear (or typical) enough to represent liver metastasis.
- The interaction between CHML and Rab14 should be further validated in HCC cell line in vivo.
- To further confirm the role of CHML in HCC metastasis, the lung metastasis should be analyzed in an orthotopic xenograft model.

Minimal concerns:

- The Rab14 protein level in YY8103 cells is not consistent in Fig5a and Fig 7c.
- In the introduction, the authors wrote that “The high mortality is attributed to high rate of recurrence after hepatic resection (HR) and radiofrequency ablation (RFA) with 5-year recurrence rates 50-100% ” Does it means that the recurrence rates of 5 year after radical treatment such as HR or RFA is nearly 100%? This is not believable, and the references are too old and not representative.

Reviewer #3 (Expertise: Protein-protein interaction, Remarks to the Author):

This is an intriguing manuscript presenting strong evidence for an oncogenic and/or metastasis-inducing role of the protein product of the CHML gene (REP2) in liver cancer. Knockdown of REP2 had no apparent influence on normal cells, presumably because REP1 can support the geranylgeranylation of Rab proteins adequately. However, REP2 KD did reduce the invasive and migratory properties of tumor cells. The authors explain this by a potentially important role of Rab14, which they found to be the predominant Rab protein in REP2 pull-down experiments. Surprisingly, the authors only identified Rab14 and smaller amount of 2 other Rabs in these experiments, but, for example, no Rab7, a prominent Rab protein. They also showed a strong interaction with Rab7 in in vitro experiments, but no interaction with Rab7. This goes against strong evidence from earlier work. Thus, the disease choroideremia (CHM) is caused by loss of REP1 activity. In CHM patients, where REP1 activity is absent, Rab7 and indeed all other essential Rabs

must be geranylgeranylated with the help of REP2, with the disease phenotype arising only after many years of normal development. In fact, REP2 has been shown to bind to Rab7 with high affinity, and the CHM phenotype has been attributed to the fact that Rab27 is underprenylated because it binds relatively weakly to REP2 and is out-competed by, for example, Rab7 (reference 30 in the submitted manuscript). The authors must address this point and provide some explanation for the discrepancies with earlier work.

The authors go on to provide support for the idea that Rab14 is involved in the invasive and migratory properties of the tumors cells by showing that Rab14 KD reduces these properties without obvious effect on non-tumor cells.

Against the background of recent evidence for an involvement of REP1 overexpression in lung, cervix and colon cancer, the main conclusions of the manuscript are potentially highly important. However, the points alluded to above must be dealt with and quite extensive modifications of style and language will be required.

Point-by-point response to referees' comments:

Reviewer #1

While the authors demonstrate that CHML is increased in HCC, they conclude that it plays a tumor promoting role. However, the increased expression may be well downstream of the actual carcinogenic event and may be a consequence of the transformation.

Response:

We agree with the reviewer that upregulation of CHML in HCC may be well downstream of the actual carcinogenic event, and we have deleted this sentence.

With respect to the cell biology experiments, I have a few concerns. To address the role of CHML in migration, it was overexpressed or knocked down in a HCC cell line which is therefor already transformed.

Response:

Considering the close relationship between CHML expression and early recurrence, MVI and PVTT, as well as its expression pattern in HCC, we supposed that CHML may enhance metastasis during HCC progression, therefore, we investigated its role in transformed HCC cells.

To address the role of CHML in cell migration in non-transformed cells, we transiently overexpressed Flag-CHML in QSG-7701 cells, a human normal liver cell line. The results revealed that CHML overexpression promoted the migratory ability of QSG-7701 cells. This

data indicates that CHML may also regulate migration in normal cells.

In addition, to confirm the metastasis-promoting function of CHML, we knocked down the expression of CHML in two more HCC cells with high metastatic capabilities: CSQT-2 and LM3. Both cells showed decreased migration and invasion upon CHML KD (Fig. 3a-d).

It is not clear why endogenous levels in Figure 2 are so low while they are high in Figure 3.

Response:

To emphasize the effect of either overexpression or knockdown, the exposure time of the bands was modulated and was different. In the revised version, we demonstrated the bands with different exposure time, thus you can find that the intensity of the bands with long exposure in Fig.2e was similar to those bands with short exposure in Fig.s3a.

With respect to the migration/invasion assays, it is not clear why controls in Figure 2 migrate so much less than the controls in Figure 3.

Response:

Similar to the above question, the incubation time of the two assays was modulated and was different. In the assay for CHML-overexpressing cells, the cells were incubated with serum-free DMEM for 5 hours, and for CHML KD cells, the incubation time was 6 hours.

In the revised manuscript, we shortened the incubation time for CHML KD and control cells to 5 hours, and updated the data (Fig.s2a-d).

Figure s2

Also, data presentation is confusing. What is "fold of changed cells" or "percentage of changed cells" in the migration/invasion assay?

Response:

We have corrected the inappropriate description, and replaced "fold of changed cells" with "relative cell number".

The co-immunoprecipitation and pull down assays provide strong evidence for an interaction between CHML and Rab14, but do not provide any insight into the domains that might mediate this interaction.

Response:

As suggested, we purified 3 truncated GST fusion protein, according to the first 2 GDI domains and the C-terminal region of CHML. The *in vitro* pull down assay demonstrated that CHML interacted with Rab14 via its 2 GDI domains (Fig.4h and i).

Further experiments test the role of CHML in Rab14 localization. While the authors state that knockdown of CHML results in a "more dispersed" distribution of Rab14 after CHML KO, the imaging presented does not support this conclusion. This is also true of Figure 6g. These data require careful quantification.

Overall, the imaging in Figure 6 is of low quality.

Response:

According to the suggestion, we have improved the quality of the images, and quantified the Rab14-nucleus distance in control and CHML KD cells (Fig. 6a and b), which revealed more dispersed localization of Rab14 upon CHML KD. We also improved the quality of Fig.6g. Due to the more dispersed localization of GDP-bound Rab14S25N, punctual staining pattern was observed and the signal was much weaker than Rab14Q70L. However, most of the punctual Rab14S25N colocalized with CHML, while rare Rab14Q70L exhibited colocalization with

CHML.

Final experiments test the types of proteins present in Rab14 positive vesicles and conclude that the presence of metastasis associated proteins provides a potential mechanism for CHML's role in cancer. However, these data highly preliminary and do not provide substantial insight. In fact, with Rab14's known localizations and function, it would be surprising if these proteins were not present.

Response:

We chose Mucin13 and CD44 from the 38 identified metastasis-related proteins in Rab14-positive vesicles for further study, based on their well-known role in metastasis. We confirmed that both proteins were in Rab14 vesicles as they partially colocalized with Rab14 in cells (Fig. 8a). To explore the influence of CHML KD on these molecules, we examined the cell surface-associated Mucin13 and CD44, and found that the membrane localization of both molecules was decreased in CHML KD cells (Fig. 8b-d). These data suggests that CHML influences HCC metastasis via regulating metastasis regulators-loaded Rab14-positive vesicles.

The manuscript needs to be carefully edited for English.

Response:

As suggested, we have carefully revised the manuscript, corrected the grammar errors and improved the writing.

Reviewer #2

Major concerns:

- The author concluded that choroideremia-like (CHML) is up-regulated in HCC and is closely associated with the metastasis of HCC. However,

the data in Table 2 and Fig. 1 is not enough to support the conclusion that “high expression of CHML predict poor survival and earlier recurrence of HCC”.

Response:

We agree with the reviewer that the clinical evidence supporting the conclusion is insufficient. In the revised manuscript, we provide more evidence. We analyzed the correlation between CHML expression and overall/relapse-free survival using our tissue microarray and a dataset from TCGA, which demonstrated the association between high CHML expression and shorter overall survival (Fig.1e, and f)/relapse-free survival (Fig.1g and h).

HCC tends to invade the intrahepatic vasculature, and both

microvascular invasion (MVI) and macrovascular invasion (portal vein tumor thrombus, PVTT) are important predictors of poor overall and recurrence-free survival in HCC. Therefore we analyze the relationship between CHML and MVI, as well as PVTT. We showed that in two independent datasets (GSE20017 and GSE9829), CHML expression is higher in HCC tissues from patients with microvascular invasion (MVI) than those without MVI (Fig. 2a). Portal vein tumour thrombus (PVTT) arise from the invasion of HCC cells into the portal vein, is a special type of intrahepatic metastasis of HCC. We found that CHML expression was gradually increased from normal tissues, primary HCC tissues, to PVTT in two independent cohorts (Fig. 2b and 2c). These data indicated the close association between increased CHML expression and HCC metastasis.

In addition, we corrected the inappropriate expression, substituting “was associated with” for “predicted”.

In addition to overall survival (OS), the association of CHML expression with the recurrence probability and time to relapse after HCC resection should be evaluated.

Response:

To analyze the relationship between CHML expression and recurrence probability and time to relapse, relapse-free survival analysis was performed using the data of our tissue microarray and the data from TCGA-LIHC. The results showed that HCC patients with high CHML expression group manifested a shorter relapse-free survival in both cohorts (Fig. 1g and h).

- The reason that 7404 and YY8103 HCC cell lines which are not commonly used were selected in this study should be explained. Besides the data of up-regulating and then knocking down the expression of CHML in 7404 and YY8103 cells, the results should be further validated in other HCC cell lines with high basic CHML levels and higher metastatic potentials.

Response:

HCC cell line 7404 and YY8103 were from Cell Bank of Type Culture Collection of Chinese Academy of Sciences, Shanghai Institute of Cell Biology, Chinese Academy of Sciences. Detailed information of these two cell lines can be found in these papers ^{1,2}. In addition, here are some studies of HCC using 7404 ³⁻⁵ or YY-8103 ⁶⁻⁸.

We have examined the expression pattern of CHML in several HCC cells (Fig. 2d). Among these cell lines, CSQT-2 is derived from PVTT tissue and is established in our laboratory ⁹, MHCC97-H and LM3 are highly metastatic HCC cell lines ^{10,11}. Result of western blot shows that CHML expression was higher in metastatic HCC cell lines (CSQT-2,

MHCC97-H and LM3) than YY8103 and 7404 cells. Therefore, we overexpressed CHML in YY8103 and 7404 cells. In addition, we knocked down (KD) the expression of CHML in CSQT-2 and LM3. In consistent with the results obtained in YY8103 and 7404 cells, CHML KD in CSQT-2 and LM3 resulted in decreased migration and invasion *in vitro* (Fig. 3a-d), as well as metastasis *in vivo* (Fig. 3e-h; Fig. s2e-h).

• Besides the effect of overexpression of CHML on cell migration and invasion of HCC cells, does CHML affect the proliferation of HCC cells?

Response:

According to the suggestion, we examined the proliferation of CHML KD cells by MTT assay, and the results demonstrated that CHML KD did not

significantly influence proliferation of HCC cells.

- In the figure 3d, the image present is not clear (or typical) enough to represent liver metastasis.

Response:

We examined the effect of CHML KD on metastasis in both intrahepatic injection and tail vein assay using HCC cells CSQT-2 and LM3 with stronger metastatic capabilities compared to YY8103 and 7404 cells. To facilitate the detection of either intrahepatic or lung metastasis, we labelled CSQT-2 cells with luciferase. We provided *in vivo* bioluminescence imaging, pictures of the resected livers, and the representative images of H&E staining (Fig. 3e-h). These data consistently revealed the suppressive effect on HCC metastasis by CHML KD.

- The interaction between CHML and Rab14 should be further validated in HCC cell line in vivo.

Response:

As suggested, we have validated the interaction between endogenous CHML and Rab14 in CSQT-2, LM3, 7404 and YY-8103 cells using an anti-CHML antibody, which could be applied in immunoprecipitation assay (Fig. 4e and f; Fig. s4c and d).

Figure s4c, d

Figure 4e, f

- To further confirm the role of CHML in HCC metastasis, the lung metastasis should be analyzed in an orthotopic xenograft model.

Response:

According to the suggestion, we chose luciferase-labelled CSQT-2 and LM3 cells to perform intrahepatic injection, based on their strong metastatic capabilities. However, although we observed the difference in

intrahepatic metastasis, we could not detect any pulmonary metastasis when we sacrificed the mice. Instead, we performed tail vein assay, and the results showed that CHML KD decreased lung colonization of HCC cells (Fig. 3g, h; Fig. s2g).

Minimal concerns:

- The Rab14 protein level in YY8103 cells is not consistent in Fig5a and Fig 7c.

Response:

To emphasize the effect of overexpression of WT Rab14 and mutants, the exposure time was modulated and was different for Fig.5a and Fig.7c. To keep consistent with Fig.5a, we have added a long exposure image to Fig. 7c.

- In the introduction, the authors wrote that “The high mortality is attributed to high rate of recurrence after hepatic resection (HR) and radiofrequency ablation (RFA) with 5-year recurrence rates 50-100% ” Does it mean that the recurrence rates of 5 year after radical treatment such as HR or RFA is nearly 100%? This is not believable, and the references are too old and not representative.

Response:

We have corrected this inappropriate expression and updated the reference.

Reviewer #3

This is an intriguing manuscript presenting strong evidence for an oncogenic and/or metastasis-inducing role of the protein product of the CHML gene (REP2) in liver cancer. Knockdown of REP2 had no apparent influence on normal cells, presumably because REP1 can support the geranylgeranylation of Rab proteins adequately. However, REP2 KD did reduce the invasive and migratory properties of tumor cells. The authors explain this by a potentially important role of Rab14, which they found to be the predominant Rab protein in REP2 pull-down experiments. Surprisingly, the authors only identified Rab14 and smaller amount of 2 other Rabs in these experiments, but, for example, no Rab7, a prominent Rab protein. They also showed a strong interaction with Rab7 in in vitro experiments, but no interaction with Rab7. This goes against strong evidence from earlier work.

Response:

In fact, we are confused when the MS/MS analysis revealed that Rab14 was the most abundant protein pulled down by Flag-CHML. To further confirm this result, we constructed a library including tens of Rab

proteins, and the immunoprecipitation assay showed the same result as the MS/MS analysis (unpublished data). Furthermore, GST pull-down assay also showed that CHML interacted with Rab14, rather than Rab7A (Fig.4g). We have repeated the above assays for several times, and consistent results were obtained. Therefore, we explored the role of CHML-Rab14 axis in HCC metastasis, rather than other Rabs.

Thus, the disease choroideremia (CHM) is caused by loss of REP1 activity. In CHM patients, where REP1 activity is absent, Rab7 and indeed all other essential Rabs must be geranylgeranylated with the help of REP2, with the disease phenotype arising only after many years of normal development.

Response:

We agree with the notion that CHML must be involved in geranylgeranylation of Rabs in CHM patients, including modification of Rab7. However, we did not detect the interaction between CHML and Rab7 in our study.

Actually, direct interaction between Rab7 and CHML is not a prerequisite for geranylgeranylation. There are two pathways for the formation of REP-Rab-RGGT ternary complex for Prenylation of Rab proteins. In the classical pathway, REP associates first with unprenylated Rab, which is then prenylated by RGGT. In the alternative pathway, REP associates first with RGGT, then this complex binds and prenylates Rab

proteins¹². Unprenylated Rab5 as well as Rab11 could not form stable complex with REP1 both *in vivo* and *in vitro*¹². Therefore, Rab7 may also be modified by CHML-RGGT in the alternative pathway, and direct interaction with CHML was not a prerequisite for it to accomplish geranylgeranylation.

In fact, REP2 has been shown to bind to Rab7 with high affinity, and the CHM phenotype has been attributed to the fact that Rab27 is underprenylated because it binds relatively weakly to REP2 and is out-competed by, for example, Rab7 (reference 30 in the submitted manuscript). The authors must address this point and provide some explanation for the discrepancies with earlier work.

Response:

Alexey et al. reported interaction between REP2 and Rab7 using fluorescence titration assay¹³, which may be more sensitive than immunoprecipitation and GST pull-down assay employed in our study. We found that Flag-tagged REP2 did not coimmunoprecipitate with Rab7 (Fig.4a and b), and GST-tagged Rab7 could not pull down His-tagged REP2 (Fig.4g). We used standard IP and GST pull-down protocols (using Triton X-100 as detergent), and we don't know whether this method is suitable for detection of REP2-Rab7 interaction. We have also constructed 3xFlag-Rab7A, and the result of immunoprecipitation was shown below, demonstrating that 3xFlag-Rab7A could not

coimmunoprecipitate with endogenous CHML. The CHML-Rab14 complex may be more stable and easier to be detected in our system. More appropriate methods are required to detect the interaction between Rab7 and CHML.

The authors go on to provide support for the idea that Rab14 is involved in the invasive and migratory properties of the tumors cells by showing that Rab14 KD reduces these properties without obvious effect on non-tumor cells.

Response:

As suggested, we have knocked down (KD) the expression of Rab14 in normal liver cell line QSG-7701, and found that downregulation of Rab14 decreased migration of QSG-7701 cells. We did not examine whether the invasion capability of QSG-7701 cells was influenced by

Rab14, since invasion is a key malignant feature of cancer cells.

Against the background of recent evidence for an involvement of REP1 overexpression in lung, cervix and colon cancer, the main conclusions of the manuscript are potentially highly important. However, the points alluded to above must be dealt with and quite extensive modifications of style and language will be required.

Response:

Accordingly, we have added contents addressing these concerns in the discussion part (page 13, 14, highlighted in yellow). In addition to address the above issue, we also made extensive modification of the style and language of the manuscript.

References

- 1 Chen, R., Zhu, D., Ye, X., Shen, D. & Lu, R. ESTABLISHMENT OF 3 HUMAN-LIVER CARCINOMA CELL-LINES AND SOME OF THEIR BIOLOGICAL CHARACTERISTICS INVITRO. *Scientia Sinica* 23, 236-251 (1980).

- 2 叶克龙 et al. 人体肝癌细胞系 YY-8103 若干生物学特征的初步观察. 上海第一医学院学报, 423-429+489-490 (1984).
- 3 Jiang, N. & Li, Y. PTOV1 Is a Novel Prognostic Marker for Hepatocellular Carcinoma Progression and Overall Patient Survival. *Gastroenterology* 148, S1022-S1023 (2015).
- 4 Zhang, X. et al. The essential role of YAP O-GlcNAcylation in high-glucose-stimulated liver tumorigenesis. *Nature Communications* 8, doi:ARTN 1528010.1038/ncomms15280 (2017).
- 5 Han, Y. M. et al. Tumor-Induced Generation of Splenic Erythroblast-like Ter-Cells Promotes Tumor Progression. *Cell* 173, 634-648, doi:10.1016/j.cell.2018.02.061 (2018).
- 6 Deng, Q. et al. E2F8 Contributes to Human Hepatocellular Carcinoma via Regulating Cell Proliferation. *Cancer Research* 70, 782-791, doi:10.1158/0008-5472.Can-09-3082 (2010).
- 7 Wang, Y. P. et al. Insulin receptor tyrosine kinase substrate activates EGFR/ERK signalling pathway and promotes cell proliferation of hepatocellular carcinoma. *Cancer Letters* 337, 96-106, doi:10.1016/j.canlet.2013.05.019 (2013).
- 8 Wu, B. H. et al. Epigenetic silencing of JMJD5 promotes the proliferation of hepatocellular carcinoma cells by down-regulating the transcription of CDKN1A. *Oncotarget* 7, 6847-6863,

doi:10.18632/oncotarget.6867 (2016).

- 9 Wang, T. et al. Characterisation of a novel cell line (CSQT-2) with high metastatic activity derived from portal vein tumour thrombus of hepatocellular carcinoma. *Br J Cancer* 102, 1618-1626, doi:10.1038/sj.bjc.6605689 (2010).
- 10 Li, Y. et al. Establishment of cell clones with different metastatic potential from the metastatic hepatocellular carcinoma cell line MHCC97. *World J Gastroenterol* 7, 630-636 (2001).
- 11 Li, Y. et al. Establishment of a hepatocellular carcinoma cell line with unique metastatic characteristics through in vivo selection and screening for metastasis-related genes through cDNA microarray. *J Cancer Res Clin Oncol* 129, 43-51, doi:10.1007/s00432-002-0396-4 (2003).
- 12 Baron, R. A. & Seabra, M. C. Rab geranylgeranylation occurs preferentially via the pre-formed REP-RGGT complex and is regulated by geranylgeranyl pyrophosphate. *Biochem J* 415, 67-75, doi:10.1042/BJ20080662 (2008).
- 13 Rak, A. et al. Structure of the Rab7 : REP-1 complex: Insights into the mechanism of rab prenylation and choroideremia disease. *Cell* 117, 749-760, doi:DOI 10.1016/j.cell.2004.05.017 (2004).

Reviewers' comments:

Reviewer #1 (Remarks to the Author):

This is a revised manuscript from Chen et al that describes a relationship between the Rab interacting protein CHML (REP2) and Rab14. The overarching goal of the work is to define the role of CHML and Rab14 in the metastasis of hepatocellular carcinoma. The clinical data as well as the in vitro and in vivo experiments show that CHML levels correlate with disease severity and impacts migration, invasion, and metastasis. The authors have largely responded to the previous criticisms. While the imaging is improved, there is no description of how the imaging was quantified in the Methods section.

Concerns relate largely to significance. There is substantial evidence in the literature that Rab14 expression is related to migration and invasion in cancer (references below). Therefore, many of the results reported here are unsurprising. While the addition of CHML as a regulator of Rab14 is new information, I am not convinced by the data showing "escort function" of CHML, leaving mechanism an open question. The fact that Rab14 partitions into membranes from a soluble pool would be expected, regardless of association with other proteins.

Rab14 Suppression Mediated by MiR-320a Inhibits Cell Proliferation, Migration and Invasion in Breast Cancer.

Yu J, Wang L, Yang H, Ding D, Zhang L, Wang J, Chen Q, Zou Q, Jin Y, Liu X. *J Cancer*. 2016 Nov 25;7(15):2317-2326. eCollection 2016.

microRNA-338-3p functions as a tumor suppressor in human non small cell lung carcinoma and targets Ras-related protein 14.

Sun J, Feng X, Gao S, Xiao Z. *Mol Med Rep*. 2015 Feb;11(2):1400-6. doi: 10.3892/mmr.2014.2880. Epub 2014 Nov 6.

PMID:25374067

A specific subset of RabGTPases controls cell surface exposure of MT1-MMP, extracellular matrix degradation and three-dimensional invasion of macrophages.

Wiesner C, El Azzouzi K, Linder S. *J Cell Sci*. 2013 Jul 1;126(Pt 13):2820-33. doi: 10.1242/jcs.122358. Epub 2013 Apr 19.

PMID: 23606746

Reviewer #2 (Remarks to the Author):

The authors have addressed most of the concerns.

Regarding "the lung metastasis should be analyzed in an orthotopic xenograft model", LM3 cell lines can be successful for lung metastasis, if they were subcutaneously implanted at first and then the tissue block was orthotopically implanted into liver.

Reviewer #3 (Remarks to the Author):

The authors have gone to considerable lengths to address the problems identified by the reviewers, but the explanation for my main objection is still not convincing. In the face of compelling evidence for a high affinity interaction between CHML and Rab7 in the literature (quoted as Alexey et al., but should be Rak et al.), the statement made in the manuscript that CHML interacts with Rab14 but not Rab7 is still very confusing. The explanation given, i.e. essentially that Rab7 does not interact with CHML but only with the complex between CHML and RabGGTase, does not hold up, since the Kd between Rab7 and CHML was found to be 5nM in the absence of RabGGTase, much stronger than Rab27, with a Kd of 800 nM. It seems to me that the only way to solve this conflict is to measure the affinity of Rab14 to CHML using the method of Rak et al., and compare it to Rab7, preferably redone just to exclude the unlikely possibility of a mistake in the earlier data. It is of course conceivable that the pulldown used only detects even higher affinity complexes, and that Rab14 belongs in this group. In this respect, it is of interest to ask whether the authors have used their pulldown assay to look at CHM (i.e. Rep1), which shows a higher affinity to Rab7 (Kd 0.5 nM) than CHML does.

Whatever the explanation is, the simple statement that CHML does not interact with Rab7 cannot be left unqualified, and the explanation given simply does not hold up.

Reviewer #1 (Remarks to the Author):

While the imaging is improved, there is no description of how the imaging was quantified in the Methods section.

Response:

As suggested, we have added the method for quantification in the Methods section (page 23).

While the addition of CHML as a regulator of Rab14 is new information, I am not convinced by the data showing "escort function" of CHML, leaving mechanism an open question. The fact that Rab14 partitions into membranes from a soluble pool would be expected, regardless of association with other proteins.

Response:

We agree with the notion that Rab14 may partition into membranes. Based on this hypothesis, CHML should also partition into membrane associated with Rab14, due to their strong interaction (kd ca. 0.36nM for Rab14-CHML interaction). However, when we quantified the WB data in Fig. 6j using imageJ, we found that the ratio of membrane fraction/supernatant fraction for CHML was much lower than Rab14. As shown below, the ratio for CHML was less than 0.5, while the value for Rab14 was five times greater. In fact, excess Flag-CHML was added to the reaction system, and not all the Flag-CHML formed a complex with Rab14. This observation indicated that CHML was functional in the process of membrane partitioning of Rab14, rather than passively partitioning into membrane with Rab14.

The different ratio was also reminiscent of the role of a GDI displacement factor (GDF)¹. The mechanism about how Rab protein dissociates from Rab-REP complex is still largely unclear, and it is possible that CHML escorts Rab14 to the membrane, where it is replaced by a GDF, leading to its decreased membrane partition after dissociation from Rab14.

1. Dirac-Svejstrup AB, Sumizawa T, Pfeffer SR. Identification of a GDI displacement factor that releases endosomal Rab GTPases from Rab-GDI. EMBO J 16, 465-472 (1997).

Reviewer #2 (Remarks to the Author):

The authors have addressed most of the concerns.

Regarding "the lung metastasis should be analyzed in an orthotopic xenograft model", LM3 cell lines can be successful for lung metastasis, if they were subcutaneously implanted first and then the tissue block was orthotopically implanted into liver.

Response:

As suggested, we have completed the orthotopic xenograft model experiment. While equal amount of tumor tissues was bound to livers of nude mice, we found that CHML KD dramatically decreased foci formation in both livers (Fig. s4a) and lungs (Fig. s4b). We also added this result in the manuscript (page 7) , and these the images below were added to Fig.s3.

Reviewer #3 (Remarks to the Author):

In the face of compelling evidence for a high affinity interaction between CHML and Rab7 in the literature (quoted as Alexey et al., but should be Rak et al.), the statement made in the manuscript that CHML interacts with Rab14 but not Rab7 is still very confusing. The explanation given, i.e. essentially that Rab7 does not interact with CHML but only with the complex between CHML and RabGGTase, does not hold up, since the Kd between Rab7 and CHML was found to be 5nM in the absence of RabGGTase, much stronger than Rab27, with a Kd of 800 nM. It seems to me that the only way to solve this conflict is to measure the affinity of Rab14 to CHML using the method of Rak et al., and compare it to Rab7, preferably redone just to exclude the unlikely possibility of a mistake in the earlier data.

Response:

As suggested, we have revised the quotation, and performed the fluorescence titration assay to determine interaction between Rab7A/Rab14 and CHML. As shown below, we found that the interaction between Rab7A and CHML was strong, with a Kd ca. 7.8 nM, however, the interaction between Rab14 and CHML was stronger, with a Kd ca. 0.36 nM. It is possible that the strong interaction between Rab14 and CHML enabled it to be easily detected by GST-pulldown assay and Co-IP analysis in our study. We added this result in the discussion part on page14.

It is of course conceivable that the pull-down used only detects even higher affinity complexes, and that Rab14 belongs in this group. In this respect, it is of interest to ask whether the authors have used their pull-down assay to look at CHM (i.e. Rep1), which shows a higher affinity to Rab7 (Kd 0.5 nM) than than CHML does.

Response:

We have performed pull-down assay using Flag-Rab7A, and as shown below, large amount of intracellular REP1 was pulled down by Rab7A. Therefore, our pull-down system seemed to detect strong interaction.

REVIEWERS' COMMENTS:

Reviewer #1 (Remarks to the Author):

As mentioned in my previous review, while I do not necessarily have substantial comments regarding the findings as they are presented, I am still not convinced that the significance of these findings warrants publication in Nature Communications. The fact that the title has a grammatical error is also galling.

Reviewer #2 (Remarks to the Author):

The authors have addressed all of my concerns.

Reviewer #3 (Remarks to the Author):

The authors have responded appropriately to my comments and I have no further objections to publication.

REVIEWERS' COMMENTS:

Reviewer #1 (Remarks to the Author):

As mentioned in my previous review, while I do not necessarily have substantial comments regarding the findings as they are presented, I am still not convinced that the significance of these findings warrants publication in Nature Communications. The fact that the title has a grammatical error is also galling.

Response: According to your suggestion, we have corrected the title as “CHML promotes liver cancer metastasis by facilitating Rab14 recycle”.

Reviewer #2 (Remarks to the Author):

The authors have addressed all of my concerns.

Response: We greatly appreciate your thoughtful comments that helped improve the manuscript.

Reviewer #3 (Remarks to the Author):

The authors have responded appropriately to my comments and I have no further objections to publication.

Response: thank you very much for the review of our manuscript.